

# Coupling of the regional climate model COSMO-CLM using OASIS3-MCT with regional ocean, land surface or global atmosphere model: description and performance

Stefan Weiher[1], Naveed Akhtar[2], Jennifer Brauch[3], Marcus Breil[4],
Edouard Davin[5], Ha T.M. Ho-Hagemann[6], Eric Maisonnave[7], Markus Thürkow[8],
and Andreas Will[1]

[1]Lehrstuhl Umweltmeteorologie, BTU Cottbus-Senftenberg
[2]Goethe-Universität Frankfurt am Main
[3]DWD Offenbach (Main)
[4]KIT Karlsruhe
[5]ETH Zürich
[6]HZG Geesthacht
[7]CERFACS, Toulouse
[8]Freie Universität Berlin

*Correspondence to:* Andreas Will, will@b-tu.de

**Abstract.** We present the prototype of a regional climate system model based on the COSMO-CLM regional climate model coupled with several model components, analyze the performance of the couplings and present a strategy to find an optimum configuration with respect to computational costs and time to solution.

The OASIS3-MCT coupler is used to couple COSMO-CLM with two land surface models (CLM and VEG3D), a regional ocean model for the Mediterranean Sea (NEMO-MED12), two ocean models for the North and Baltic Sea (NEMO-NORDIC and TRIMNP+CICE) and the atmospheric component of an earth system model (MPI-ESM). We present a unified OASIS3-MCT interface which handles all couplings in a similar way, minimizes the model source code modifications and describes

the physics and numerics of the couplings. Furthermore, we discuss solutions for specific regional coupling problems like handling of different domains, multiple usage of MCT interpolation library and efficient exchange of 3D fields.

A series of real-case simulations over Europe has been conducted and the computational performance of the couplings has been analyzed. The usage of the LUCIA tool of the OASIS3-MCT

coupler enabled separation of the direct costs of: coupling, load imbalance and additional computations. The resulting limits for time to solution and costs are shown and the potential of further improvement of the computational efficiency is summarized for each coupling.

It was found that the OASIS3-MCT coupler keeps the direct coupling costs of communication and horizontal interpolation small in comparison with the costs of the additional computations and



load imbalance for all investigated couplings. For the first time this could be demonstrated for an exchange of approximately 450 2D fields per time step necessary for the atmosphere-atmosphere coupling between COSMO-CLM and MPI-ESM.

A procedure for finding an optimum configuration for each of the couplings was developed considering the time to solution and costs of the simulations. The optimum configurations are presented

for sequential and concurrent coupling layouts. The procedure applied can be regarded as independent on the specific coupling layout and coupling details.

## 1 Introduction

Most of the current Regional Climate Models (RCMs) lack frameworks for the interactivity between the atmosphere and the other components of the climate system. The interactivity is either altered

by the use of a simplified component model (e.g. over land) or even partly suppressed when top and lateral and/or ocean surface boundary conditions of the atmospheric model are prescribed by reanalysis or large-scale Earth System Model (ESM) outputs.

The neglected meso-scale feedbacks and inconsistencies of the boundary conditions (Laprise et al., 2008; Becker et al., 2015) might be well accountable for a substantial part of large- and

regional-scale biases found in RCM simulations at 10–50 km horizontal resolution (see e.g. Kotlarski et al. (2014) for Europe). This hypothesis gains further evidence from the results of convection-permitting simulations, in which these processes are not regarded either. These simulations provide more regional-scale information and improve e.g. the precipitation distribution in mountainous regions but they usually do not show a reduction of the large-scale biases (see e.g. Prein et al. (2013)).

The potential of explicit simulation of the processes neglected or prescribed in these land-atmosphere RCMs has been investigated using ESMs with variable horizontal resolution (Hertwig et al., 2015; Hagos et al., 2013), RCMs two-way coupled with the atmospheric component of global ESMs (Lorenz and Jacob, 2005; Inatsu and Kimoto, 2009), two-way coupled with regional oceans (Döscher et al., 2002; Gualdi et al., 2013; Zou and Zhou, 2013; Bülow et al., 2014; Akhtar et al., 2014; Pham

et al., 2014; Ho-Hagemann et al., 2013, 2015) and/or with more sophisticated land surface models (Wilhelm et al., 2014; Davin et al., 2011).

Besides various improvements, a significant increase of climate change signal was found by Somot et al. (2008) in the ARPEGE model with the horizontal grid refined over Europe and two-way coupled with a regional ocean for the Mediterranean Sea. These results strongly suggest that building

Regional Climate System Models (RCSMs) with explicit modeling of the interaction between meso scales in the atmosphere, ocean and land-surface, with large scales in the atmosphere (and ocean) is necessary to consistently represent regional climate dynamics and gain further insights into regional climate change.



The non-hydrostatic regional climate model COSMO-CLM (Rockel et al., 2008) belongs to the
class of land-atmosphere RCMs that do not allow a meso-scale interaction between different compo-
nents of the climate system. In this paper we present a first step of a coupling appoach which aims at
overcoming the previously mentioned deficiencies - individual two-way coupling of COSMO-CLM
with other climate component models using OASIS3-MCT (Valcke, 2013) over Europe. These cli-
mate component models are the Community Land Model (CLM) version 4.0, the soil and vegetation
model VEG3D for the land component, the NEMO model version 3.2 for the Mediterranean, the re-
gional ocean model TRIMNP along with the sea ice model CICE and the NEMO model (version 3.3,
including the LIM3 sea ice model) for the Baltic and the North Sea and, finally, the global Earth Sys-
tem Model MPI-ESM for the large-scale global atmosphere. Additional model components, which
are not discussed in this article but can be coupled with COSMO-CLM via OASIS3-MCT are the
ocean model ROMS (Byrne et al., 2015) and the hydrological model ParFLOW (Gasper et al., 2014)
together with CLM.

An alternative coupling strategy available for COSMO-CLM is based on an internal coupling of
the models of interest with the master routine MESSy resulting in the compilation of one executable
(Kerkweg and Joeckel, 2012). This coupling strategy is not investigated in this study.

The coupled climate models, either global (ESMs) or regional (RCSMs), are obviously computa-
tionally demanding. This is not only due to the sum of the costs of the individual model components,
but also additional costs of the coupler, additional computations needed for coupling, load imbal-
ances and/or inappropriate numerical properties of the coupled model components. Maintaining a
reasonable computational cost contributes to a large extent to models' usability. For this reason the
present paper also focuses on the coupled systems' computational efficiency which greatly relies on
the parallelization of the OASIS3-MCT coupler.

Optimization of the computational performance is considered to be highly dependent on the model
system and/or the computational machine used. However, several studies show transferability of op-
timization strategies and universality of certain aspects of the performance. Worley et al. (2011)
analyzed the performance of the Community Earth System Model (CESM) and found a good scala-
bility of the concurrently running CLM and sequentially running CICE down to approximately 100
grid points per processor for two different resolutions and computing architectures. Furthermore,
they found the CICE scalability to be limited by a domain decomposition, which follows that of
the ocean model, resulting to a very low number of ice grid points in subdomains. Lin-Jiong et al.
(2012) investigated a weak scaling (discussed in section 4.2) of the FAMIL model (IAP, Beijing)
and found a performance similar to that of the optimized configuration of the CESM (Worley et al.,
2011). This result indicates that a careful investigation of the model performance leads to similar
results for similar computational problems. An analysis of CESM at very high resolutions by Den-
nis et al. (2012) showed that a cost reduction by a factor of three or less can be achieved using an
optimal layout of model components. Later Alexeev et al. (2014) presented an algorithm for finding



an optimum model coupling layout (concurrent, sequential) and processor distribution between the model components minimizing the load imbalance in CESM.

These results indicate that the optimized computational performance is weakly dependent on the computing architecture or on the individual model components but on the coupling method. Further-

more, the application of an optimization procedure was found beneficial.

In this study we present a detailed analysis of coupled COSMO-CLM performances on the IBM POWER6 machine *Blizzard* located at DKRZ, Hamburg. We calculate the speed and costs of the individual model components and of the coupler itself and identify the causes of reduced speed or increased costs for each coupling and reasonable processor configurations. We suggest an optimum

configuration for different couplings considering costs and speed of the simulation and discuss the current and potential performances of the coupled systems. Particularities of the performance of a coupled RCM are highlighted together with the potential of the new coupling software OASIS3-MCT. We suggest a procedure of optimization of an RCSM, which can be generalized. However, we will show that some relevant optimizations are possible only due to features available with the

OASIS3-MCT coupler.

The paper is organized as follows: The coupled model components are described in section 2. Section 3 focuses on the OASIS3-MCT coupling method and its interfaces for the individual couplings. The coupling method description encompasses the OASIS3-MCT functionality, method of the coupling optimization and particularities of coupling of a regional climate model system. The

model interface description gives a summary of the physics and numerics of the individual couplings. In section 4 the computational efficiency of individual couplings is presented and discussed. Finally, the conclusions and an outlook are given in section 5. For improved readability Tables 1 and 2 provide an overview of the acronyms frequently used throughout the paper and of the investigated couplings.

**2   Description of model components**

The further development of the COSMO model in Climate Mode (COSMO-CLM) presented here aims at overcoming the limitations of the regional soil-atmosphere climate model, as discussed in the introduction, by replacing prescribed vegetation, lower boundary condition over sea surfaces and the lateral and top boundary conditions with interactions between dynamical models.

The models selected for coupling with COSMO-CLM need to fulfill the requirements of the intended range of application which are (1) the simulation at varying scales from convection-resolving up-to-50 km grid spacing, (2) local-scale up to continental-scale simulation domains and (3) full capability at least for European model domains. We decided to couple the NEMO ocean model for the Mediterranean Sea (NEMO-MED12) and the Baltic and Northern Seas (NEMO-NORDIC), al-

ternatively the TRIMNP regional ocean model together with the sea ice model CICE for the Baltic



and Northern Seas (TRIMNP+CICE), the Community Land Model (CLM) of soil and vegetation (replacing the multi-layer soil model TERRA), alternatively the VEG3D soil and vegetation model and the global Earth System Model MPI-ESM for two-way coupling with the regional atmosphere. Table 2 gives an overview of all coupled-model systems investigated, their components and the in-
stitutions at which they are maintained. An overview of the coupled models selected for coupling with COSMO-CLM (CCLM) is given in table 3 together with some key aspects of the configuration used in this study.

In the following, the model components used are briefly described with respect to model history, space-time scales of applicability and model physics and dynamics relevant for the coupling.

## 2.1 COSMO-CLM

COSMO-CLM is the COSMO model in climate mode. The COSMO model is a non-hydrostatic limited-area atmosphere-soil model originally developed by Deutscher Wetterdienst for operational numerical weather prediction (NWP). Additionally, it is used for climate, environmental (Vogel et al., 2009) and idealized studies (Baldauf et al., 2011).
The COSMO physics and dynamics are designed for operational applications at horizontal resolutions of 1 to 50 km for NWP and RCM applications. The basis of this capability is a stable and efficient solution of the non-hydrostatic system of equations for the moist, deep atmosphere on a spherical, rotated, terrain-following, staggered Arakawa C grid with a hybrid z-level coordinate. The model physics and dynamics are discribed in Doms et al. (2011) and Doms and Baldauf (2015)
respectively. The features of the model are discussed in Baldauf et al. (2011).

The model's climate mode (CLM) (Rockel et al., 2008) is a technical extension for long-time simulations and all related developments are unified with COSMO regularly. The important aspects of the climate mode are time dependency of the vegetation parameters and of the prescribed SSTs and usability of the output of several global and regional climate models as initial and boundary
conditions. All other aspects related to CLM e.g. the restart option for soil and atmosphere, the NetCDF model in- and output, online computation of climate quantities, and the sea ice module or spectral nudging can be used in other modes of the COSMO model as well.

The model version `cosmo_4.8_clm19` is the recommended version of the CLM-Community (Kotlarski et al., 2014) and it is used as basis of the development of the couplings.

## 2.2 MPI-ESM

The global Earth System Model of the Max Planck Institute for Meteorology Hamburg (MPI-ESM; Stevens et al. (2013)) consists of subsystem models for ocean, atmo-, cryo-, pedo- and the bio-sphere. The hydrostatic general circulation model ECHAM6 uses the transform method for horizontal computations. The derivatives are computed in spectral space, while the transports and physics
tendencies on a regular grid in physical space. A pressure-based sigma coordinate is used for vertical



discretization. The ocean model MPIOM (Jungclaus et al., 2013) is a regular grid model with the option of local grid refinement. The terrestrial bio- and pedo-sphere component model is JSBACH (Reick et al., 2013; Schneck et al., 2013). The marine biogeochemistry model used is HAMOCC5 (Ilyina et al., 2013). A key aspect is the implementation of the bio-geo-chemistry of the carbon

cycle, which allows e.g. investigation of the dynamics of the greenhouse gas concentrations (Giorgetta et al., 2013). The subsystem models are coupled via the OASIS3-MCT coupler (Valcke, 2013) which was implemented recently by I. Fast of DKRZ in the CMIP5 model version. This allows parallelized and efficient coupling of a huge amount of data, is a requirement of atmosphere-atmosphere coupling.

The reference MPI-ESM configuration uses a spectral resolution of T63, which is equivalent to a spatial resolution of about 320 km for atmospheric dynamics and 200 km for model physics. Vertically the atmosphere is resolved by 47 hybrid sigma-pressure levels with the top level at 0.01 hPa. The reference MPIOM configuration uses the GR15L40 resolution which corresponds to a bipolar grid with a horizontal resolution of approximately 165 km near the Equator and 40 vertical levels,

most of them within the upper 400 m. The North and the South Pole are located over Greenland and Antarctica in order to avoid the "pole problem" and to achieve a higher resolution in the Atlantic region (Jungclaus et al., 2013).

### 2.3  NEMO

The Nucleus for European Modelling of the Ocean (NEMO) is based on the primitive equations.
It can be adapted for regional and global applications. The sea ice (LIM3) or the marine biogeochemistry module with passive tracers (TOP) can be used optionally. NEMO uses staggered variable positions together with a geographic or Mercator horizontal grid and a terrain-following $\sigma$-coordinte (curvilinear grid) or a z-coordinate with full or partial bathymetry steps (orthogonal grid). A hybrid vertical coordinate (z-coordinate near the top and $\sigma$-coordinate near the bottom boundary) is possible
as well (for details see Madec (2011)).

The coupling of COSMO-CLM with the global ocean model NEMO is realized by means of two different regional versions of the NEMO model adapted to specific conditions of the region of application. For the North and Baltic Seas, the sea ice module (LIM3) of NEMO is activated and the model is applied with a free surface to enable the tidal forcing. Whereas in the Mediterranean
Sea, the ocean model runs with a classical rigid-lid formulation in which the sea surface height is simulated via pressure differences. Both model setups are briefly introduced in the following two sub-sections.

#### 2.3.1  Mediterranean Sea

Lebeaupin et al. (2011), Beuvier et al. (2012) and Akhtar et al. (2014) adapted the NEMO version 3.2
(Madec, 2008) to the regional ocean conditions of the Mediterranean Sea, hereafter called *NEMO-*



*MED12*. It covers the whole Mediterranean Sea excluding the Black Sea. The NEMO-MED12 grid is a section of the standard irregular ORCA12 grid (Madec, 2008) with an eddy-resolving 1/12° horizontal resolution, stretched in latitudinal direction, equivalent to 6–8 km horizontal resolution. In the vertical, 50 unevenly spaced levels are used with 23 levels in the top layer of 100 m depth. A
time step of 12 min is used.

The initial conditions for potential temperature and salinity are taken from the Medatlas (MEDAR-Group, 2002). The fresh-water inflow from rivers is prescribed by a climatology taken from the RivDis database (Vörösmarty et al., 1996) with seasonal variations calibrated for each river by Beuvier et al. (2010) based on Ludwig et al. (2009). In this context, the Black Sea is considered as a river
for which climatological monthly values are calculated from a dataset of Stanev and Peneva (2002). The water exchange with the Atlantic Ocean is parameterized using a buffer zone west of the Strait of Gibraltar with a thermohaline relaxation to the World Ocean Atlas data of Levitus et al. (2005).

### 2.3.2   North and Baltic Seas

Hordoir et al. (2013), Dieterich et al. (2013) and Pham et al. (2014) adapted the NEMO version 3.3
to the regional ocean conditions of the North and Baltic Sea, hereafter called *NEMO-NORDIC*. Part of NEMO 3.3 is the sea ice model LIM3 including a representation of dynamic and thermodynamic processes (for details see Vancoppenolle et al. (2009)). The NEMO-NORDIC domain covers the whole Baltic and North Sea with two open boundaries to the Atlantic Ocean: the southern, meridional boundary in the English Channel and the northern, zonal boundary between the Hebride Islands and
Norway. The horizontal resolution is 2 nautical miles (about 3.7 km) with 56 stretched vertical levels. The time step used is 5 min. No fresh-water flux correction for the ocean surface is applied. NEMO-NORDIC uses a free top surface to include the tidal forcing in the dynamics. Thus, the tidal potential has to be prescribed at the open boundaries in the North Sea. Here, we use the output of the global tidal model of Egbert and Erofeeva (2002).

The lateral fresh-water inflow from rivers plays a crucial role for the salinity budget of the North and Baltic Seas. It is taken from the daily time series of river runoff from the E-HYPE model output operated at SMHI (Lindström et al., 2010). The World Ocean Atlas data (Levitus et al., 2005) are used for the initial and lateral boundary conditions of potential temperature and salinity.

### 2.4   TRIMNP and CICE

TRIMNP (Tidal, Residual, Intertidal Mudflat Model Nested Parallel Processing) is the regional ocean model of the University of Trento, Italy (Casulli and Cattani, 1994; Casulli and Stelling, 1998). The domain of TRIMNP covers the Baltic Sea, the North Sea and a part of the North East Atlantic Ocean with the north-west corner over Iceland and the south-west corner over Spain at the Bay of Biscay. TRIMNP is designed with a horizontal grid mesh size of 12.8 km and 50 vertical layers.
The thickness of the top 20 layers is each 1 m and increases with depth up to 600 m for the remaining





layers . The model time step is 240 s. Initial states and boundary conditions of water temperature, salinity, and velocity components for the ocean layers are determined using the monthly ORAS-4 reanalysis data of ECMWF (Balmaseda et al., 2013). The daily Advanced Very High Resolution Radiometer AVHRR2 data of the National Oceanic and Atmospheric Administration of USA are used for surface temperature and the World Ocean Atlas data (Levitus and Boyer, 1994) for surface salinity. No tide is taken into account in the current version of TRIMNP. The climatological means of fresh-water inflow of 33 rivers to the North Sea and the Baltic Sea are collected from Wikipedia.

The sea ice model CICE version 5.0 is developed at the Los Alamos National Laboratory, USA (http://oceans11.lanl.gov/trac/CICE/wiki), to represent dynamic and thermodynamic processes of sea ice in global climate models (for more details see Hunke et al. (2013)). In this study CICE is adapted to the region of the Baltic Sea and Kattegat, a part of the North Sea, on a 12.8 km grid with five ice categories. Initial conditions of CICE are determined using the AVHRR2 SST.

### 2.5 VEG3D

VEG3D is a multi-layer soil-vegetation-atmosphere transfer model (Schädler, 1990) designed for regional climate applications and maintained by the Institute of Meteorology and Climate Research at the Karlsruhe Institute of Technology. VEG3D considers radiation interactions with vegetation and soil, calculates the turbulent heat fluxes between the soil, the vegetation and the atmosphere, as well as the thermal transport and hydrological processes in soil, snow and canopy.

The radiation interaction, the moisture and turbulent fluxes between soil surface and the atmosphere are regulated by a massless vegetation layer located between the lowest atmospheric level and the soil surface, having its own canopy temperature, specific humidity and energy balance. The multi-layer soil model solves the heat conduction equation for temperature and the Richardson equation for soil water content. Thereby, vertically differing soil types can be considered within one soil column, comprising 10 stretched layers with its bottom at a depth of 15.34 m. The heat conductivity depends on the soil type and the water content. In case of soil freezing the ice-phase is taken into account. The soil texture has 17 classes. Three classes are reserved for water, rock and ice. The remaining 14 classes are taken from the USDA Textural Soil Classification (Staff, 1999).

Ten different landuse classes are considered: water, bare soil, urban area and seven vegetation types. Vegetation parameters like the leaf area index or the plant cover follow a prescribed annual cycle.

Up to two additional snow layers on top are created, if the snow cover is higher than 0.01 m. The physical properties of the snow depend on its age, its metamorphosis, melting and freezing. A snow layer on a vegetated grid cell changes the vegetation albedo, emissivity and turbulent transfer coefficients for heat as well.

An evaluation of VEG3D in comparison with TERRA in West Africa is presented by Köhler et al. (2012).



### 2.6 Community Land Model

The Community Land Model (CLM) is a state-of-the-art land surface model designed for climate applications. Biogeophysical processes represented by CLM include radiation interactions with veg-
etation and soil, the fluxes of momentum, sensible and latent heat from vegetation and soil and the heat transfer in soil and snow. Snow and canopy hydrology, stomatal physiology and photosynthesis are modeled as well.

Subgrid-scale surface heterogeneity is represented using a tile approach allowing five different land units (vegetated, urban, lake, glacier, wetland). The vegetated land unit is itself subdivided into
17 different plant-functional types (or more when the crop module is active). Temperature, energy and water fluxes are determined separately for the canopy layer and the soil. This allows a more realistic representation of canopy effects than by bulk schemes, which have a single surface temperature and energy balance. The soil column has 15 layers, the deepest layer reaching 42 meters depth. Thermal calculations explicitly account for the effect of soil texture (vertically varying), soil liquid
water, soil ice and freezing/melting. CLM includes a prognostic water table depth and groundwater reservoir allowing for a dynamic bottom boundary conditions for hydrological calculations rather than a free drainage condition. A snow model with up to five layers enables the representation of snow accumulation and compaction, melt/freeze cycles in the snow pack and the effect of snow aging on surface albedo.

CLM also includes processes such as carbon and nitrogen dynamics, biogenic emissions, crop dynamics, transient land cover change and ecosystem dynamics. These processes are activated optionally and are not considered in the present study. A full description of the model equations and input datasets is provided in Oleson et al. (2010) (for `CLM4.0`) and Oleson et al. (2013) (for `CLM4.5`). An offline evaluation of `CLM4.0` surface fluxes and hydrology at the global scale is provided by
Lawrence et al. (2011).

CLM is developed as part of the Community Earth System Model (CESM) (Collins et al., 2006; Dickinson et al., 2006) but it has been also coupled to other global (NorES) or regional (Steiner et al., 2005, 2009; Kumar et al., 2008) climate models. In particular, an earlier version of CLM (`CLM3.5`) has been coupled to COSMO (Davin et al., 2011; Davin and Seneviratne, 2012) using
a "sub-routine" approach for the coupling. Here we use a more recent version of CLM (`CLM4.0` as part of the `CESM1_2.0` package) coupled to COSMO via OASIS3-MCT rather than through a sub-routine call. Note that `CLM4.5` is also included in `CESM1_2.0` and can be also coupled to COSMO using the same framework.

### 3 Description and optimization of COSMO-CLM couplings via OASIS3-MCT

The computational performance, usability and maintainability of a complex model system depend on the coupling method used, the ability of the coupler to run efficiently in the computing architecture,





and on the flexibility of the coupler to deal with different requirements on the coupling depending on model physics and numerics.

In the following, the physics and numerics of the coupling of COSMO-CLM with the different
model components via OASIS3-MCT are discussed and the different aspects of optimization of the computational performance of the individual couplings are highlighted. In section 3.1 the main properties of the OASIS3-MCT coupling method are described, the new OASIS3-MCT features highlighted and the steps of optimization of the computational performance are described. In sections 3.2 to 3.5 the physics and numerics of the couplings are described. In these sections a list of the
exchanged variables, the additional computations and the interpolation methods are presented. The time step organization of each coupled model is given in the Appendix A.

### 3.1   OASIS3-MCT coupling method and performance optimization

Lateral-, top- and/or bottom-boundary conditions for regional geophysical models are traditionally read from files and updated regularly at runtime. We call this approach *offline (one-way) coupling*.
For various reasons, one could decide to calculate these boundary conditions with another geophysical model - at runtime - in an *online (one-way) coupling*. If this additional model in return receives information from the first model modifying the boundary conditions provided by the first to the second, an *online two-way coupling* is established. In any of these cases, model exchanges must be synchronized. This could be done by (1) reading data from file, (2) calling one model as a subroutine
of the other or (3) by using a coupler which is a software that enables online data exchanges between models.

Communicating information from model to model boundaries via reading from and writing to a file is known to be quite simple to implement but computationally inefficient, particularly in the case of non-parallelized I/O and high frequencies of disc access. In contrast, calling component
models as COSMO-CLM subroutines exhibits much better performances because the information is exchanged directly in memory. Nevertheless, the inclusion of an additional model in a "subroutine style" requires comprehensive modifications of the source code. Furthermore, the modifications need to be updated for every new source code version. Since the early 90s, software solutions have been developed, which allow coupling between geophysical models in a non-intrusive, flexible and
computationally efficient way.

One of the software solutions for coupling of geophysical models is the OASIS coupler, which is widely used in the climate modeling community (see for example Valcke (2013) and Maisonnave et al. (2013)). Its latest fully parallelized version, OASIS3-MCT version 2.0 (Valcke et al., 2013), proved its efficiency for high-resolution quasi-global models on top-end supercomputers (Masson
et al., 2012).

In the OASIS coupling paradigm, each model is a *component* of a *coupled system*. Each component is included as a separate executable.



### 3.1.1 The OASIS3-MCT coupling method

OASIS3-MCT consists of a FORTRAN Application Programming Interface (API), whose subrou-
tines have to be added in all coupled-system components. The part of the program in which the
OASIS3-MCT API routines are located is called *component interface*. There is no independent OA-
SIS executable anymore, as was the case with OASIS3. With OASIS3-MCT every communication
between the model components is directly executed via the Message Passing Interface (MPI) li-
brary. This significantly improves the performance over OASIS3, because the bottleneck due to the
sequential separate coupler is entirely removed as shown e. g. in Gasper et al. (2014).

In the following, we point out the potential of the new OASIS3-MCT coupler and discuss the
peculiarities of its application for coupling in the COSMO model in CLimate Mode (COSMO-
CLM). If there is no difference between the OASIS versions, we use the acronym OASIS, otherwise
the OASIS version is specified.

At runtime, all components are launched together on a single MPI context. The parameters defin-
ing the properties of a coupled system are provided to OASIS via an ASCII file called *namcouple*.
By means of this file the components, coupling fields and coupling intervals are associated. Specific
calls of the *OASIS3-MCT Application Programming Interface (API)* in a *component interface* de-
scribed in sections 3.2 to 3.5 define a component's coupling characteristics, that is, (1) the name of
incoming and outgoing coupling fields, (2) the grids on which each of the coupling fields are dis-
cretized, (3) a mask (binary-sparse array) describing where coupling fields are described on the grids
and (4) the partitioning (MPI-parallel decomposition into subdomains) of the grids. This component
partitioning does not have to be the same for each component as OASIS3-MCT is able to scatter and
gather the arrays of coupling fields if they are exchanged with a component model that is decom-
posed differently. Similarly, OASIS is able to perform interpolations between different grids. OASIS
also is able to perform time averages for exchanges at a coupling time step, e. g. if the components'
time steps differ. In total, six to eight API routines have to be called by each component model to
start MPI communications, declare the component's name, possibly get back MPI local commu-
nicator for internal communications, declare the grid partitioning and variable names, finalize the
component's coupling characteristics declaration, send and receive the coupling fields and, finally,
close the MPI context at the component's runtime end. The number of routines, whose arguments
require easily identifiable model quantities, is the most important feature of the OASIS3-MCT cou-
pling library that contributes to its non-intrusiveness. In addition, each component can be modified
separately or another component can be added later. This facilitates a shared maintenance between
the users of the coupled-model system: when a new development or a version upgrade is done in one
component, the modification scarcely affects the other components. This ensures the modularity and
interoperability of any OASIS-coupled system.

As previously mentioned, OASIS3-MCT includes the MPI library for direct parallel communi-
cations between components. To ensure that calculations are delayed only by receiving of coupling



375 fields or interpolation of these fields, MPI non-blocking sending is used by OASIS3-MCT so that sending coupling fields is a quasi-instantaneous operation. The SCRIP library (Jones, 1997) included in OASIS3-MCT provides a set of standard operations (for example bilinear and bicubic interpolation, Gaussian-weighted N-nearest-neighbor averages) to calculate, for each source grid point, an interpolation weight that is used to derive an interpolated value at each (non-masked) target grid

380 point. OASIS3-MCT can also (re-)use interpolation weights calculated offline. Intensively tested for demanding configurations (Craig et al., 2011), the MCT library performs the definition of the parallel communication pattern needed to optimize exchanges of coupling fields between each component's MPI subdomain. It is important to note that unlike the "subroutine coupling" each component coupled via OASIS3-MCT can keep its parallel decomposition so that each of them can be used at its

385 optimum scalability. In some cases, this optimum can be adjusted to ensure a good load balance between components. The two optimization aims that strongly matter for computational performance are discussed in the next section.

### 3.1.2 The coupled-system synchronization and optimization

A coupled model component receiving information from one or several other components has to wait

390 for the information before it can perform its own calculations. In case of a two-way coupling this component provides information needed by the other coupled-system component(s). As mentioned earlier, the information exchange is quasi-instantaneously performed, if the time needed to perform interpolations can be neglected which is the case even for 3D-field couplings (as discussed in section 4.5.1). Therefore, the total duration of a coupled-system simulation can be separated into two parts

395 for each component: (1) a *waiting time* in which a component waits for boundary conditions and (2) a *computing time* in which a component's calculations are performed. The duration of a stand-alone, that is, un-coupled component simulation approximates the coupled-component's computing time. In a coupled system this time can be shorter than in the uncoupled mode, since the reading of boundary conditions from file (in stand-alone mode) is partially or entirely replaced by the coupling. It is

400 also important to note that components can perform their calculations *sequentially* or *concurrently*.

The coupled-system's total sequential simulation time can be expected to be equal to the sum of the individual component's calculation times, potentially increased by the time needed to interpolate and communicate coupling fields between the components. The computational constraint induced by a sequential coupling algorithm depends on the computing architecture. If one process can be started

405 on each core, the cores allocated for one model component are idle while others are performing calculations and vice versa. In such a case the performance optimization strategy needs to consider model component waiting time. If more than one process can be started on each core, each model component can use all cores sequentially and an allocation of the same number of cores to each model component can avoid any waiting time. This is discussed in more detail in the following

410 paragraphs.



The constraints of sequential coupling are often alleviated if calculations of a coupled-system component can be performed with coupling fields of another component's previous coupling time step. This concurrent coupling strategy is possible if one of the two sets of exchanged quantities is slowly changing in comparison to the other set. For example, sea surface temperatures of an ocean model are slowly changing in comparison to fluxes coming from an atmosphere model. However, now the time to solution of each model component can be substantially different and an optimisation strategy needs to minimize the waiting time.

Thus, the strategy of synchronization of the model components depends on the layout of the coupling (sequential or concurrent) in order to reduce the waiting time as much as possible. It is important to note that huge differences in computational performance can be found for different coupling layouts due to different scalability of the modular model components.

Since computational efficiency is one of the key aspects of any coupled system the various aspects affecting it are discussed. These are the performances of the model components, of the coupling library and of the coupled system. Hereby the design of the interface and the OASIS3-MCT coupling parameters, which enables optimization of the efficiency, are described.

The model component performance depends on the component's scalability. The optimum partitioning has to be set for each parallel component by means of a strong scaling analysis (discussed in section 4.1. This analysis, which results in finding the scalability limit (the maximum speed) or the scalability optimum (the acceptable level of parallel efficiency), can be difficult to obtain for each component in a multi-component context. In this article, we propose to simply consider the previously defined concept of the computing time (excluding the waiting time from the total time to solution). In chapter 4 we will describe our strategy to separate the measurement of computing and waiting times for each component and how to deduce the optimum MPI partitioning from the scaling analysis.

The optimization of OASIS3-MCT coupling library performance is relevant for the efficiency of the data exchange between components discretized on different grids. The parallelized interpolations are performed by the OASIS3-MCT library routines called by the source or by the target component. An interpolation will be faster if performed (1) by the model with the larger number of MPI processes available (up to the OASIS3-MCT interpolation scalability limit) and/or (2) by the fastest model (until the OASIS3-MCT interpolation together with the fastest model's calculations last longer than the calculations of the slowest model).

A significant improvement of interpolation performance can be achieved by coupling of multiple variables that share the same coupling characteristics via a single communication, that is, by using the technique called *pseudo-3D coupling*. Via this option, a single interpolation and a single send/receive instruction are executed for a whole group of coupling fields, for example, all levels and variables in an atmosphere-atmosphere coupling at one time instead of all coupling fields and levels separately. The option groups several small MPI messages into a big one and, thus, reduces commu-





nications. Furthermore, the amount of matrix multiplications is reduced because it is performed on big arrays. This functionality can easily be set via the 'namcouple' parameter file (see section B.2.4 in Valcke et al. (2013)). The impact on the performance of COSMO-CLM atmosphere-atmosphere coupling is discussed in section 4.5.1). See also Maisonnave et al. (2013).

The optimization of the performance of a coupled system relies on the allocation of an optimum number of computing resources to each model. If the components' calculations are performed concurrently the waiting time needs to be minimized. This can be achieved by balancing the load of the two (or more) components between the available computing resources: the slower component is granted more resources leading to an increase in its parallelism and a decrease in its computing time. The opposite is done for the fastest component until an equilibrium is reached. Chapter 4 gives examples of this operation and describes the strategy to find a compromize between each component's optimum scalability and the load balance between all components.

On all high-performance operating systems it is possible to run one process of a parallel application on one core in a so-called *single-threading* (ST) mode (fig. 1a). Should the core of the operating system feature the so-called *simultaneous multi-threading* (SMT) mode, two (or more) processes/threads of the same (in a *non-alternating processes distribution* (fig.1b)) or of different (in an *alternating processes distribution* (fig.1c)) applications can be executed simultaneously on the same core. Applying SMT mode is more efficient for well-scaling parallel applications leading to an increase in speed in the order of magnitude of 10 % compared to the ST mode. Usually it is possible to specify, which process is executed on which core (see fig. 1). In this cases the SMT mode with alternating distribution of model component processes can be used, and the waiting time of sequentially coupled components can be avoided. Starting each model component on each core is usually the optimum configuration, since the reduction of waiting time of cores outperforms the increase of the time to solution by using ST mode instead of SMT mode (at each time one process is executed on each core). In the case of concurrent couplings, however, it is possible to use SMT mode with a non-alternating processes distribution.

The optimization procedure applied is described in more detail in section 4.2 for the couplings considered. The results are discussed in section 4.5.1.

### 3.1.3 Regional climate model coupling particularities

In addition to the standard OASIS functionalities, some adaptation of the OASIS3-MCT API routines were necessary to fit special requirements of the regional-to-regional and regional-to-global couplings presented in this article.

A regional model covers only a portion of earth's sphere and requires boundary conditions at its domain boundaries. This has two immediate consequences for coupling: first, two regional models do not necessarily cover exactly the same part of earth's sphere. This implies that the geographic boundaries of the model's computational domains and of coupled variables may not be the same in



the source and target components of a coupled system. Second, a regional model can be coupled with
a global model or another limited-area model and some of the variables which need to be exchanged
are three-dimensional as in the case of atmosphere-to-atmosphere or ocean-to-ocean coupling.

A major part of the OASIS community uses global models. Therefore, OASIS standard features
fit global model coupling requirements. Consequently, the coupling library must be adapted or used
in an unconventional way, described in the following, to be able to cope with the extra demands
mentioned.

Limited-area field exchange has to deal with a mismatch of the domains of the coupled model
components. Differences between the (land and ocean) models coupled to COSMO-CLM lead to two
solutions for the mismatch of the model domains. For coupling with the Community Land Model
(CLM) the CLM domain is extended in such a way that at least all land points of the COSMO-CLM
domain are covered. Then, all CLM grid points located outside of the COSMO-CLM domain are
masked. To achieve this, a uniform array on the COSMO-CLM grid is interpolated by OASIS3-
MCT to the CLM grid using the same interpolation method as for the coupling fields. On the CLM
grid the uniform array contains the projection weights of the COSMO-CLM on the CLM grid points.
This field is used to construct a new CLM domain containing all grid points necessary for interpola-
tion. However, this solution is not applicable to all coupled-system components. In ocean models, a
domain modification would complicate the definition of ocean boundary conditions or even lead to
numerical instabilities at the new boundaries. Thus, the original ocean domain, that must be smaller
than the COSMO-CLM domain, is interpolated to the COSMO-CLM grid. At runtime, all COSMO-
CLM ocean grid points located inside the interpolated area are filled with values interpolated from
the ocean model and all COSMO-CLM ocean grid points located outside the interpolated area are
filled with external forcing data.

Multiple usage of the MCT library occured in the CCLM+CLM coupled system implementation
making some modifications of the OASIS3-MCT version 2.0 necessary. Since the MCT library has
no re-entrancy properties, a duplication of the MCT library and a renaming of the OASIS3-MCT
calling instruction were necessary. This modification ensures the capability of coupling any other
CESM component via OASIS3-MCT. The additional usage of the MCT library occured in the CESM
framework of CLM version 4.0. More precisely, the DATM model interface in the CESM module is
using the CPL7 coupler including the MCT library for data exchange.

Interpolation of 3D fields is necessary in an atmosphere-to-atmosphere coupling. The OASIS3-
MCT library is used to provide 3D boundary conditions to the regional model and a 3D feedback
to the global coarse-grid model. OASIS is not able to interpolate the 3D fields vertically, mainly
because of the complexity of vertical interpolations in geophysical models (different orographies,
level numbers and formulations of the vertical grid). However, it is possible to decompose the oper-
ation into two steps: (1) horizontal interpolation with OASIS3-MCT and (2) model-specific vertical
interpolation performed in the source or target component's interface. The first operation does not



require any adaption of the OASIS3-MCT library and can be solved in the most efficient manner by the pseudo-3D coupling option described in section 3.1.2. The second operation requires a case-dependent algorithm addressing aspects such as inter- and extra-polation of the boundary layer over different orographies, change of the coordinate variable, conservation properties as well as interpo-

lation efficiency and accuracy.

An exchange of 3D fields, which occurs in the CCLM+MPI-ESM coupling, requires a more intensive usage of the OASIS3-MCT library functionalities than observed so far in the climate modeling community. The 3D regional-to-global coupling is even more computationally demanding than its global-to-regional opposite. Now, all grid points of the COSMO-CLM domain have to be interpo-

lated instead of just the grid points of a global domain that are covered by the regional domain. The amount of data exchanged is rarely reached by any other coupled system of the community due to (1) the high number of exchanged 2D fields, (2) the high number of exchanged grid points (full COSMO-CLM domain) and (3) the high exchange frequency at every ECHAM time step. In addition, as will be explained in section 3.2, the coupling between COSMO-CLM and MPI-ESM needs

to be sequential and, thus, the exchange speed has a direct impact on the simulation's total time to solution.

Interpolation methods used in OASIS3-MCT are the SCRIP standard interpolations: bilinear, bicubic, first- and second-order conservative. However, the interpolation accuracy might not be sufficient and/or the method is inappropriate for certain applications. This is for example the case with the

atmosphere-to-atmosphere coupling CCLM+MPI-ESM. The linear methods turned out to be of low accuracy and the second-order conservative method requires the availability of the spatial derivatives on the source grid. Up to now, the latter cannot be calculated efficiently in ECHAM (see section 3.2 for details). Other higher-order interpolation methods can be applied by providing weights of the source grid points at the target grid points. This method was successfully applied in the CCLM+MPI-

ESM coupling by application of a bicubic interpolation using a 16-point stencil. In section 3.2 to 3.5 the interpolation methods recommended for the individual couplings are given.

### 3.2 CCLM+MPI-ESM

In the CCLM+MPIESM two-way coupled system the 3D atmospheric fields are exchanged between the atmospheres of COSMO-CLM and MPI-ESM running sequentially. In MPI-ESM the COSMO-

CLM tendencies can be regarded as a parameterization of meso-scale processes in a limited domain of the global atmosphere. In COSMO-CLM the MPI-ESM boundary conditions are used as in standard one-way nesting. Both atmosphere models run sequentially.

COSMO-CLM recalculates the ECHAM time step in dependence on the lateral- and top-boundary conditions provided by ECHAM. In ECHAM the solution is updated in a limited area of the globe

using the solution provided by COSMO-CLM. For computational-efficiency reasons the data exchange in ECHAM is done in grid point space. This avoids costly transformations between grid



point and spectral space. Since the simulation results of COSMO-CLM need to become effective in ECHAM dynamics, the two-way coupling is implemented in ECHAM after the transformation from spectral to grid point space and before the computation of advection (see Fig. 9 and DKRZ (1993)

for details).

ECHAM provides the boundary conditions for COSMO-CLM at time level $t = t_n$ of the three time levels $t_n - (\Delta t)_E$, $t_n$ and $t_n + (\Delta t)_E$ of ECHAM's leap frog time integration scheme. However, the second part of the Assilin time filtering in ECHAM for this time level has to be executed after the advection calculation in `dyn` (see Fig. 9) in which the tendency due to two-way coupling needs to be

included. Thus, the fields sent to COSMO-CLM as boundary conditions do not undergo the second part of the Assilin time filtering. The COSMO-CLM is integrated over j time steps between the ECHAM time level $t_{n-1}$ and $t_n$. However, the coupling time may also be a multiple of an ECHAM time step.

A complete list of variables exchanged between ECHAM and COSMO-CLM is given in Table 4.

The data sent by ECHAM are the 3D variables of COSMO-CLM temperature, u- and v-components of the wind velocity, specific humidity, cloud liquid and ice water content and the two-dimensional fields surface pressure, surface temperature and surface snow amount. At initial time the surface geopotential is sent to COSMO-CLM for calculation of the orography differences between the model grids. After horizontal interpolation to the COSMO-CLM grid via the bilinear SCRIP interpolation[1]

the 3D variables are vertically interpolated to the COSMO-CLM grid keeping the height of the 300 hPa level constant and using the hydrostatic approximation. Afterwards, the horizontal wind vector velocity components of ECHAM are rotated from the geographical (lon, lat) ECHAM to the rotated (rlon, rlat) COSMO-CLM coordinate system. Here `send_fld` ends and the interpolated data are used to initialize the boundlines at next COSMO-CLM time levels $t_m = t_{n-1} + k \cdot (\Delta t)_C \leq$

$t_n$, with $k \leq j = (\Delta t)_E/(\Delta t)_C$. However, the final time of COSMO-CLM integration $t_{m+j} = t_m + j \cdot (\Delta t)_C = t_n$ is equal to the time $t_n$ of the ECHAM data received.

After integrating between $t_n - i \cdot (\Delta t)_E$ and $t_n$ the 3D fields of temperature, u- and v velocity components, specific humidity and cloud liquid and ice water content of COSMO-CLM are vertically interpolated to the ECHAM vertical grid following the same procedure as in the COSMO-CLM

receive-interface and keeping the height of the 300 hPa level of the COSMO-CLM pressure constant. The wind velocity vector components are rotated back to the geographical directions of the ECHAM grid. The 3D fields and the hydrostatically approximated surface pressure are sent to ECHAM, horizontally interpolated to the ECHAM grid by OASIS3-MCT[2] and received in ECHAM grid space. In ECHAM the COSMO-CLM solution is relaxed at the lateral and top boundaries of the COSMO-

---

[1]This interpolation is used for the performance tests only. For physical coupling the conservative interpolation second order (CO2) is used, which requires the additional computation of derivatives. Alternatively, a bicubic interpolation can be used that has the same accuracy as CO2.

[2]The bilinear interpolation is used. The usage of a second-order conservative interpolation requires horizontal derivatives of the variables exchanged. This is not implemented in this version of the COSMO-CLM send interface.



CLM domain by means of a cosine weight function over a range of five to ten ECHAM grid boxes using a weight between zero at the outer boundary and one in the central part of the COSMO-CLM domain. Additional fields are calculated and relaxed in the COSMO-CLM domain for a consistent update of the ECHAM prognostic variables. These are the horizontal derivatives of temperature, surface pressure, u and v wind velocity, divergence and vorticity.

The two-way coupled system CCLM+MPI-ESM with prescribed COSMO-CLM solution within the COSMO-CLM domain (weight=1) provides a stable solution over climatological time scales. A strong initialization perturbation is avoided by slowly increasing the maximum coupling weight to 1 with time.

### 3.3  CCLM+NEMO-MED12

COSMO-CLM and the NEMO ocean model are coupled concurrently for the Mediterranean Sea (NEMO-MED12) and for the North and Baltic Sea (NEMO-NORDIC). Table 5 gives an overview of the variables exchanged. Bicubic interpolation between the horizontal grids is used for all variables.

At the beginning of the NEMO time integration (see Fig. 8) the COSMO-CLM receives the sea surface temperature (SST) and - only in the case of coupling with the North and Baltic Sea - also

the sea ice fraction from the ocean model. At the end of each NEMO time step COSMO-CLM sends average water, heat and momentum fluxes to OASIS3-MCT. In the NEMO-NORDIC setup COSMO-CLM additionally sends the averaged sea level pressure (SLP) needed in NEMO to link the exchange of water between North and Baltic Sea directly to the atmospheric pressure. The sea ice fraction affects the radiative and turbulent fluxes due to different albedo and roughness length of

ice. In both coupling setups SST is the lower boundary condition for COSMO-CLM and it is used to calculate the heat budget in the lowest atmospheric layer. The averaged wind stress is a direct momentum flux for NEMO to calculate the water motion. Solar and non-solar radiation are needed by NEMO to calculate the heat fluxes. $E - P$ ("Evaporation minus Precipitation") is the net gain ($E - P > 0$) or loss ($E - P < 0$) of fresh water at the water surface. This water flux adjusts the

salinity of the uppermost ocean layer.

In all COSMO-CLM grid cells where there is no active ocean model underneath, the lower boundary condition (SST) is taken from ERA-Interim re-analyses. The sea ice fraction in the Atlantic Ocean is derived from the ERA-Interim SST where $SST < -1.7\,^{\circ}C$ which is a salinity-dependent freezing temperature.

On the NEMO side, the coupling interface is included similar to COSMO-CLM, as can be seen in Fig. 10. There is a setup of the coupling interface at the beginning of the NEMO simulation. At the beginning of the time loop NEMO receives the upper boundary conditions from OASIS3-MCT and before the time loop ends, it sends the coupling fields (average SST and sea ice fraction for NEMO-NORDIC) to OASIS3-MCT.



## 3.4 CCLM+TRIMNP+CICE

In the CCLM+TRIMNP+CICE coupled system (denoted as COSTRICE; Ho-Hagemann et al. (2013)), all fields are exchanged every hour between the three models COSMO-CLM, TRIMNP and CICE running concurrently. An overview of variables exchanged among the three models is given in Table 5. As shown in Fig. 8, COSMO-CLM receives the skin temperature ($T_{Skin}$) at the beginning of each COSMO-CLM time step over the coupling areas, the North and Baltic Seas. The skin temperature $T_{skin}$ is a weighted average of sea ice and sea surface temperature. It is not a linear combination of skin temperatures over water and over ice weighted by the sea ice fraction. Instead, the skin temperature over ice $T_{Ice}$ and the sea ice fraction $A_{Ice}$ of CICE are sent to TRIMNP where they are used to compute the heat flux $HFL$, that is, the net outgoing long-wave radiation. $HFL$ is used to compute the skin temperature of each grid cell via the Stefan-Boltzmann Law.

At the end of the time step, after the physics and dynamics computations and output writing, COSMO-CLM sends the variables listed in Table 5 to TRIMNP and CICE for calculation of wind stress, fresh water, momentum and heat flux. TRIMNP can either directly use the sensible and latent heat fluxes from COSMO-CLM (considered as flux coupling method; see e.g. Döscher et al. (2002)) or compute the turbulent fluxes using the temperature and humidity density differences between air and sea as well as the wind speed (considered as the coupling method via state variables; see e.g. Rummukainen et al. (2001)). The method used is specified in the subroutine `heat_flux` of TRIMNP.

In addition to the fields received from COSMO-CLM, the sea ice model CICE requires from TRIMNP the SST, salinity, water velocity components, ocean surface slope, and freezing/melting potential energy. CICE sends to TRIMNP the water and ice temperature, sea ice fraction, fresh-water flux, ice-to-ocean heat flux, short-wave flux through ice to ocean and ice stress components. The horizontal interpolation method applied in CCLM+TRIMNP+CICE is the SCRIP nearest-neighbour inverse-distance-weighting fourth-order interpolation (DISTWGT).

## 3.5 CCLM+VEG3D and CCLM+CLM

The two-way coupling between COSMO-CLM and the land surface models VEG3D or CLM is similar to the other in several respects. First, the call to the LSM (OASIS send and receive; see Fig. 8) is placed at the same location in the code as the call to COSMO-CLM's native land surface scheme, TERRA_ML, which is switched off when either VEG3D or CLM is used. This ensures that the sequence of calls in COSMO-CLM remains the same regardless of whether TERRA_ML, VEG3D or CLM is used. In the default configuration used here COSMO-CLM and CLM (or VEG3D) are executed sequentially, thus mimicking the "subroutine"-type of coupling used with TERRA_ML. Note that it is also possible to run COSMO-CLM and the LSM concurrently but this is not discussed





here. Details of the time step organization of VEG3D and CLM are described in the appendix and
shown in Fig. 13 and 14 .

VEG3D runs at the same time step and on the same horizontal rotated grid ( 0.44° here) as
COSMO-CLM with no need for any horizontal interpolations. CLM uses a regular lat-lon grid and
the coupling fields are interpolated using bilinear interpolation (atm to LSM) and distance-weighted
interpolation (LSM to atm). The time step of CLM is synchronized with the COSMO-CLM radia-
tive transfer scheme time step (one hour in this application) with the idea that the frenquency of the
radiation update determines the radiative forcing at the surface.

The LSMs need to receive the following atmospheric forcing fields (see also Table 6): the total
amount of precipitation, the short- and long-wave downward radiation, the surface pressure, the wind
speed, the temperature and the specific humidity of the lowest atmospheric model layer.

CLM additionally receives the atmospheric forcing height[3] for calculation of turbulence in the
atmospheric boundary layer. VEG3D additionally needs infomation about the time-dependent com-
position of the vegetation to describe its influence on radiation interactions and turbulent fluxes cor-
rectly. This includes the leaf area index, the plant cover and a vegetation function which describes
the annual cycle of vegetation parameters based on a simple cosine function depending on latitude
and day. They are exchanged at the beginning of each simulated day.

One specificity of the coupling concerns the turbulent fluxes of latent and sensible heat. In its tur-
bulence scheme, COSMO-CLM does not directly use surface fluxes. It uses surface states (surface
temperature and humidity) together with turbulent diffusion coefficients of heat, moisture and mo-
mentum. Therefore, the diffusion coefficients need to be calculated from the surface fluxes received
by COSMO-CLM. This is done by deriving, in a first step, the coefficient for heat (assumed to be the
same as the one for moisture in COSMO-CLM) based on the sensible heat flux. In a second step an
effective surface humidity is calculated using the latent heat flux and the derived diffusion coefficient
for heat.

## 4    Computational efficiency

Optimising the performance of a coupled model system can save a substantial amount of resources
in terms of simulation time or costs. Sometimes, it is even a prerequisite for the applicability of a
model system at higher resolutions or on climatological time scales. There are two main goals of a
performance analysis: (1) To identify code patterns of inefficient behaviour in parallel applications
for a given resources configuration by using sophisticated tools such as e.g. SCALASCA (Geimer
et al., 2010) and VampirTrace (Müller et al., 2008). (2) To analyze the scalability of a coupled model
system and its components in order to obtain an optimum configuration of resources. The second is
the subject of this chapter. For this purpose the *Load-balancing Utility and Coupling Implementation*

---

[3]This field is needed for initialization only. In this test series it is exchanged at every coupling time.





*Appraisel* (LUCIA), developed at CERFACS, Toulouse, France (Maisonnave and Caubel, 2014) is used, which is available together with the OASIS3-MCT coupler.

More precisely, we investigate the scalability of each coupled system's components in terms of simulation speed, computational costs and parallel efficiency, the time needed for horizontal interpolations by OASIS3-MCT and the load balance in the case of concurrently running components. Based on these results, an optimum configuration for all couplings is suggested. Finally, the costs of the optimium configurations are compared with an optimum stand-alone COSMO-CLM configura-
tion and the potential for further optimization is discussed.

### 4.1 Simulations setup and methodology

A parallel program's runtime $T(n, R)$ mainly depends on two variables: the problem size $n$ and the number of cores $R$, that is, the resources. In scaling theory, a *weak scaling* is performed with the notion to solve an increasing problem size in the same time, while as in a *strong scaling* a fixed
problem size is solved more quickly with an increasing amount of resources. Due to resources limits on the common high-performance computer we chose to conduct a strong-scaling analysis with a common model setup allowing for an easier comparability of the results. By means of the scalability study we identified an optimum configuration for each coupling which served as basis to address two central questions: (1) How much does it cost to add one (or more) component(s) to COSMO-CLM?
(2) How big are the costs of OASIS3-MCT to transform the information between the components' grids? The first question can only be answered by a comparison to a reference which is, in this study, a stand-alone COSMO-CLM simulation. The second question can directly be answered by the measurements of LUCIA. We used this part of the OASIS3-MCT library to measure the computing and waiting time of each component in a coupled model system (see section 3.1.2) as well as the
time needed for interpolation of fields before and after sending or receiving.

A common model setup for the CORDEX-EU domain was chosen for the reference model COSMO-CLM. The other components' setups are those used by the developers of the particular coupling (see section 2). The simulated period is one month, the horizontal grid has 132 by 129 grid points and $0.44°$ (ca. 50 km) horizontal grid spacing. In the vertical, 45 levels are used for the CCLM+MPI-
ESM and CCLM+VEG3D couplings as well as for the stand-alone COSMO-CLM simulations. All other couplings use 40 levels. The impact of this difference on the numerical performance is compansated by a simple post-processing scaling of the measured COSMO-CLM computing time $T_{CCLM,45}$ of the COSMO-CLM components that employ 45 levels assuming a linear scaling of the COSMO-CLM computing time with the number of levels as [4] $T_{CCLM} = 0.8 \cdot T_{CCLM,45} \cdot \frac{40}{45} + 0.2 \cdot$
$T_{CCLM,45}$ The usage of a real-case configuration allows to provide realistic computing times.

---

[4]The estimation that 80 % of COSMO-CLM's computations depend on the number of model levels is based on COSMO-CLM's internal time measurements. $T_{CCLM,45}$ is the time measured by LUCIA.



The computing architecture used is *Blizzard* at *Deutsches Klimarechenzentrum* (DKRZ) in Hamburg, Germany. It is an IBM Power6 machine with nodes consisting of 16 dual-core CPUs (16 processors, 32 cores). A simultaneous multi-threading (SMT; see section 3.1.2) allows to launch two processes on each core. A maximum of 64 threads that can be launched on one node.

The measures used in this paper to present and discuss the computational performance are well known in scalability analyses: (1) *time to solution* in Hours Per Simulated Year (HPSY), (2) *costs* in Core Hours Per Simulated Year (CHPSY) and (3) *parallel efficiency* (PE) (see Table 7 for details).

Usually, $HPSY_1$ is the time to solution of a model component executed serially, that is, using one process ($R = 1$) and $HPSY_2$ is the time to solution if executed using $R_2 > R_1$ parallel processes.

Some model components, like ECHAM, cannot be executed serially. This is why the reference number of threads is $R_1 \geq 2$ for all coupled-system components.

In a perfectly scaling parallel application the costs would remain constant if the resources are doubled, the parallel efficiency would be 100 %, the speed would be doubled and the speed-up would be 200 %. A parallel efficiency of 50 % is reached if the costs $CHPSY_2$ are twice as big as those of

the reference configuration $CHPSY_1$.

### 4.2   Strategy for finding an optimum configuration

The optimization strategy that we pursue is rather empirical than strictly mathematical, which is why we understand "optimum" more as "near-optimum". Nonetheless, our results show that these empirical methods are sufficient for the complexity of the couplings investigated here and lead to satisfying

results. Besides costs and time to solution, we suggest a limit for parallel efficiency of 50 % until which increasing costs can be regarded as still acceptable. Usually, this is limiting the time to solution which can be achieved and depends on the cost-efficiency of the reference configuration. In this study for all couplings the one-node configuration is regarded to have 100 % parallel efficiency. This leads to the constraint $R_{CCLM} = R_{CClm+CLM} = R_{CCLM+VEG3D} = R_{CCLM+ECHAM+MPIOM} = \#nodes\cdot$

32. for the number $R$ of cores investigated, and a clear strategy for finding the maximum number of nodes for which $PE \geq 50\%$.

The strategies for identifying an optimum configuration are different for sequential and concurrent couplings due to the possible waiting time which needs to be considered with concurrent couplings.

For sequentially running components (CCLM+CLM and CCLM+MPI-ESM) we used the SMT

mode and an alternating distribution of processes to make sure that all cores were busy at all times. Hereby possible component-internal load imbalances due to e.g. parts of the code not executed in parallel are neglected. A detailed analysis of CCLM+MPI-ESM performance on one node ($n = 1$) showed a significant reduction of time to solution and costs, if alternating instead of non-alternating distribution of processes in SMT mode (see section 4.5.1 for details) is used.

The optimum configuration is found by starting the measuring of the computing time on one node for all components, doubling the resources and measuring the computing time again and again as long




as each component's gain in speed, compared to its speed on one node, outweighs the increase in costs. If costs are, however, not an issue it is suggested to stop increasing resources before a parallel efficiency of 50 % of each component model is reached.

For concurrent couplings (CCLM+NEMO-MED12 and CCLM+TRIMNP+CICE) the SMT mode with non-alternating process distribution is used aiming to speed up all components in comparison to the ST mode. The constraint for the distribution of cores is $\sum_{m=1}^{M} R_m = \#nodes \cdot 32$. A summary of the configuration of each coupled system is given in Table 8.

The optimization process of a concurrently coupled model system additionally needs to consider minimising the load imbalance between all components. This means that the computing times of all components need to be similar in order to reduce the costs due to idle cores. Practically speaking, one starts with a first-guess distribution of processes between all components on one node, measure each component's computing and waiting time and adjust the processes distribution between the model components if the waiting time of at least one component is larger than 5 % of the total runtime.

If, finally, the waiting times of all components are small, the following chain of action is repeated several times: doubling resources for each component, measuring computing times, adjusting and re-distributing the processes if necessary. If costs are a limiting factor this is repeated until the costs reach a pre-defined limit. If costs are not a limiting factor, the procedure should be repeated until the model with the highest time to solution reaches the proposed parallel-efficiency limit of 50 %.

### 4.3 Scalability results

Figure 2 shows the results of the performance measurement *time to solution* for all model components individually in coupled mode and for stand-alone COSMO-CLM (in ST and SMT mode). As reference, the slopes of a model at no speed-up and at perfect speed-up are shown. Three groups can be identified. CLM and VEG3D have the shortest times to solution and, thus, they are the fastest components. The three ocean components and the COSMO-CLM components in coupled as well as in stand-alone mode need about 2–10 HPSY. The overall slowest components are CICE and ECHAM which need about 20 HPSY independently on the amount of resources used. Within the range of resources investigated CICE, ECHAM and VEG3D exhibit almost no speed-up. On the contrary, MPIOM, NEMO-MED12 and CLM have a very good scalability up to the tested limit of 128 cores.

Figure 3 shows the second relevant performance measure, the absolute costs of computation in CPUh per simulated year for the same couplings together with the perfect and no speed-up slopes. The afore mentioned three groups slightly change their composition. VEG3D and CLM are not only the fastest but also the cheapest components, the latter becoming even cheaper with increasing resources. A little bit more expensive but mostly in the same order of magnitude as the land surface components are the ocean components MPIOM and TRIMNP followed by CICE, NEMO-MED12 and all the different COSMO-CLM components. The NEMO model is approximately two times more expensive than TRIMNP. Surprisingly, the CICE model is as expensive as the regional climate





model COSMO-CLM. The most expensive coupled component is ECHAM with almost doubled costs as resources are doubled. This and the high coupling costs of COSMO-CLM coupled to MPI-
ESM will be analyzed in section 4.5.1 and 4.5.2.

In order to analyze the performance of the couplings in more detail we took measurements of stand-alone COSMO-CLM in single-threading (ST) and multi-threading (SMT) mode. The direct comparison provides the information of how much COSMO-CLM's speed benefits from switching from ST to SMT mode. As shown in Fig. 2 at 16 cores the COSMO-CLM in SMT mode is 27 %
faster. When allocating 128 cores both modes arrive at about the same speed and costs. The parallel efficiency shown in Fig. 4 allows to understand this behavior. COSMO-CLM in ST and SMT mode exhibits a very similar PE for the same number of processes and an increased loss of PE between 160 and 80 grid points per process. This can be explained by a weak scalability of unavoidable communication of data between the threads computing the values in subdomains. The values at
three grid points close to the subdomain boundary need to be communicated to the thread computing the values in the neighboring grid points. In conclusion, it is recommended to keep the number of horizontal grid points per process higher than $100 = 10 \times 10$.

The difference in time to solution (Fig. 2) and costs (Fig. 3) between coupled and stand-alone COSMO-CLM is a direct measure of the additional time to solution and costs due to the COSMO-
CLM component interface. Hereby, the number of cores and the threading mode (ST or SMT) are kept constant. COSMO-CLM components of concurrent couplings are compared to stand-alone COSMO-CLM in SMT mode. COSMO-CLM components of sequential couplings are compared to stand-alone COSMO-CLM in ST mode. The latter has the same amount of processes per node and only one process per core. For coupling COSMO-CLM to ocean models NEMO-MED12 and
TRIMNP+CICE, these additional times to solution and costs are 1–5 % at 16 cores and 5–13 % at 32 cores. The comparison of coupled and stand-alone COSMO-CLM in ST mode at 32 cores exhibits 11 % addtional time to solution and costs for COSMO-CLM coupled to VEG3D and 76 % for COSMO-CLM coupled to MPI-ESM. At 128 cores, the differences increase to 21 and 93 % respectively. It is worth noting here that COSMO-CLM coupled to CLM should exhibit about the same
coupling costs as COSMO-CLM coupled to VEG3D since both coupling interfaces lead to similar times to solution. However, as mentioned in section 2.6 CLM is coupled to `cosmo_5.0_clm1` model version which is a more recent version than `cosmo_4.8_clm19` used for all other couplings presented here. Therefore, the true additional costs can be slightly different.

The parallel efficiency shown in Fig. 4 gives a better understanding of the development of costs
and speed. For CLM it exhibits a so-called *super-linear speed-up* which has not been investigated in detail. The components CICE, ECHAM and VEG3D exhibit a very fast loss of PE close to the no-speed-up limit indicating nearly no scalability. TRIMNP looses PE fast in comparison to NEMO-MED12 indicating "no speed-up" of some parts of the model. The ocean models MPIOM and NEMO-MED12 are still far away from the PE limit.



### 4.4 The optimum configurations


Based on the results of the scalability study, we recommend an optimum configuration for stand-alone COSMO-CLM and all coupled systems which are summarized in Fig. 5 and Table 8. Considering time to solution and costs, we find that the optimum processes configuration for stand-alone COSMO-CLM is 64 cores using SMT mode resulting in 3.6 HPSY and costs of 230.4 CHPSY. This

configuration will be used as common reference for all couplings to quantify the additional time and costs of adding one or more components to COSMO-CLM.

The optimum configurations of the couplings with CLM and VEG3D are identical: the coupled system is using SMT mode and 128 cores for each component model. In both couplings, the time to solution of the coupled land-surface component is small in comparison to COSMO-CLM. CLM

needs only 22 % of VEG3D's time to solution. The different COSMO-CLM version used in the coupling CCLM+CLM has a longer time to solution and costs and a higher parallel efficiency. That's why the gain in speed still dominates the increase in costs at 128 cores compared to the measurements at 32 cores. In the CCLM+VEG3D coupling the weak scaling behavior of VEG3D can be neglected because COSMO-CLM dominates the coupled system's costs. At 128 cores, COSMO-CLM used

in the coupling CCLM+VEG3D reaches a point at which the increase in costs slightly dominates the gain in speed. From this perspective, running on 96 cores would be preferable. We nonetheless chose 128 cores for a better comparison to CCLM+CLM. Both coupled-system's time to solution is only marginally bigger than that of stand-alone COSMO-CLM: 4.0 HPSY for CCLM+CLM and 3.7 HPSY for CCLM+VEG3D. The corresponding costs are about double the costs of the stand-

alone reference: 512.0 and 473.6 CHPSY, respectively. The costs of the OASIS3-MCT interpolations are 3.0 % of the total coupled-system's CHPSY in the CCLM+CLM coupling which is still acceptable. There are no interpolations performed for CCLM+VEG3D.

NEMO-MED12 scales very well in the analyzed resources range making COSMO-CLM the limiting component of the CCLM+NEMO-MED12 coupled system. Because the load imbalance was

unacceptably high at a resources distribution of 64 by 64 cores, it was decided to run NEMO-MED12 with 14 cores less and giving these to COSMO-CLM resulting in an overall decrease in load imbalance to an acceptable 3.9 % of the total costs. Surprisingly, increasing the number of cores for COSMO-CLM did not change much the time to solution. The corresponding NEMO-MED12 measurements at 50 cores are a bit out of scaling as well. This is probably caused by the I/O which

increased for unknown reasons on the machine used between the time of conduction of the first series of simulations and of the optimized simulations. A further increase in resources is not recommended because COSMO-CLM already approached the parallel-efficiency limit by using 78 cores. The coupled systems's optimum time to solution and costs are 4.0 HPSY and 512.0 CHPSY, respectively. The costs for OASIS3-MCT interpolations are negligible with 0.03 % of the total costs.

Due to CICE's low speed-up and the fact that the time to solution of CICE is generally one order of magnitude higher than that of TRIMNP and COSMO-CLM, there is no common speed of all





three components. Clearly, CICE is the limiting component in this coupled system so that more than 32 cores altogether can not be used efficiently. Considering CICE's parallel efficiency, more than 10 cores are not feasible dividing up the rest into 16 for COSMO-CLM and 6 for TRIMNP in the optimum configuration. The total time to solution is 18.0 HPSY and the total costs amount to 576.0 CHPSY of which 20.9 % are wasted in load imbalance.

In the CCLM+MPI-ESM coupling, ECHAM is the limiting component model making it not feasible for the coupled system to run on more than 32 cores. This configuration leads to a total time of solution of 34.8 HPSY and total costs of 1113.6 CHPSY of which 3.6 % are due to the load imbalance between MPIOM and ECHAM. The costs of OASIS3-MCT horizontal interpolations are considerably small with 0.7 % of the total costs.

### 4.5 Extra time and costs

Figure 5 exhibits significant differences between the times to solution (vertical axis) and costs (box area) of the model components at optimum configurations of the coupled model systems and the COSMO-CLM stand-alone time to solution and costs. These results are given quantitatively in the columns of Table 8. Its first section summarizes the configuration of each coupling. The second section gives the absolute and relative time to solution of the coupled systems together with the relative difference between *time to solution* for the coupled system and COSMO-CLM stand-alone ($CCLM_{sa}$), given as $CS - CCLM_{sa}$. In the following section the absolute and relative costs are given follwed by relative extra costs of OASIS3-MCT horizontal interpolation and of the load imbalance. Finally, the relative differences of *costs* are given between the coupled system and COSMO-CLM stand-alone ($CS - CCLM_{sa}$), between the coupled and stand-alone COSMO-CLM ($CCLM - CCLM_{sa}$) and between the coupled and stand-alone COSMO-CLM using the same resources as COSMO-CLM in the coupled mode ($CCLM - CCLM_{sa,sc}$). The relative extra time and costs are given in % of the reference $CCLM_{sa}$ time to solution and costs, respectively.

The CCLM+VEG3D coupling can be identified as the coupling with the smallest extra time (2.8 %) and extra costs (105.6 %). The coupling CCLM+CLM is just slightly more expensive with 11.1 % additional time and 122.2 % additional costs. However, the couplings with soil-vegetation models do not need to have extra costs. In this case the coupled model is replacing TERRA, which is the internal soil-vegetation model of COSMO-CLM. All other couplings need to simulate additionally the regional ocean or global earth system dynamics.

The coupling with the Mediterranean Sea (CCLM+NEMO-MED12) is as expensive as CCLM+CLM. The coupling with the North and Baltic Sea (CCLM+TRIMNP+CICE) takes 3.5 times longer due to a lack of scalability of the sea ice model CICE and costs 1.5 times more than the optimum stand-alone COSMO-CLM. The most expensive coupling presented here is the coupling with the global atmosphere (CCLM+MPI-ESM). It takes 7.5 times longer due to lack of scalability of the additional computations in MPI-ESM and costs almost four times more. Section 4.5.2, in which the





CCLM+MPI-ESM extra time and costs are discussed, provides a comparison with MPI-ESM stand-alone as well.

The comparison of costs of the coupled and stand-alone COSMO-CLM (Table 8 line 14) shows a major dependency on the number of allocated cores. Despite the longer runtime, COSMO-CLM coupled to TRIMNP+CICE is by 27 % cheaper than the optimum stand-alone COSMO-CLM only because of 16 cores used instead of 64. The additional costs of COSMO-CLM using 78 cores and coupled to NEMO-MED12 are 35.4 %, of COSMO-CLM using 128 cores and coupled to VEG3D

and CLM are 87.2 % and 119.2 %, respectively. An exception are the additional costs of 83.1 % for COSMO-CLM using 32 cores and coupled to MPI-ESM.

     To quantify the additional costs by the COSMO-CLM coupling interface, all coupled COSMO-CLM components are compared to the stand-alone COSMO-CLM reference using the same configuration (thread mode and number of cores; see Table 8, line 15). The COSMO-CLM interface with the

smallest additional costs of 4.9 % is the one of COSMO-CLM coupled to NEMO-MED12, followed by 17.2 % when coupled with TRIMNP+CICE, 20.4 % when coupled to VEG3D. The additional costs of COSMO-CLM coupled to CLM are 40.9 %. However, they are not the true additional costs due to different COSMO-CLM versions used in stand-alone and in the coupled-system simulations. The coupling interface of COSMO-CLM coupled to MPI-ESM exhibits the biggest additional costs

with 76.4 % (see section 4.5.2 for details).

     Figure 5 shows no direct coupling costs of the OASIS3-MCT coupler. This is due to the fact that they are negligible in comparison with the costs of the model components. This is not necessarily the case, in particular when a huge amount of fields is exchanged. The relevant steps to reduce the direct coupling costs are described in section 4.5.1.

The extra costs of coupling given in Table 8 for CCLM as $CCLM - CCLM_{sa,sc}$ are resulting from additional computations necessary for coupling. They are described in section 4.5.2.

### 4.5.1 Direct coupling costs

The CCLM+MPI-ESM coupling is one of the most intensive couplings that has up to now been realized with OASIS3(-MCT) in terms of number of coupling fields and coupling time steps: 450

2D fields are exchanged every ECHAM coupling time step, that is, every ten minutes (see section 3.2). Most of these 2D fields are levels of 3D atmopsheric fields. We show in this section that a conscious choice of coupling software and computing platform features can have a significant impact on simulation speed and costs.

     To make the CCLM+MPI-ESM coupling more efficient, all levels of a 3D variable are sent and

received in one MPI message using the concept of *pseudo-3D coupling* as described in section 3.1.2, reducing the number of sent and received fields (see Table 4). The change from 2D to pseudo-3D coupling lead to a decrease of the costs of the coupled system by 3.7 %, which corresponds to 35,7 % of $CCLM_{sa}$. Since this measured computing time does not include OASIS3-MCT interpolations,





the decrease can be attributed to a reduction in MPI communications. The costs of the OASIS3-MCT

interpolations are reduced to 24.0 % which corresponds to an overall additional reduction of 1.4 %

of the costs of the coupled system or 13.5 % of $CCLM_{sa}$.

The second optimization step is a change of hardware usage. Instead of non-alternating, an alternating processes distribution of cores is used. On one node, this reduced the coupled system's time to solution and costs by 35.1 %. An even higher decrease was found for MPI-ESM due to a dramatic

reduction of the time to solution of the inefficient calculation of the derivatives (needed for coupling with COSMO-CLM only) by one process. The COSMO-CLM's time to solution in coupled mode was reduced by 9.2 %. This gain is smaller than what could have been expected from the stand-alone COSMO-CLM measurements. Going from 16 cores in SMT mode to 32 cores in ST mode is resulting in a reduction of time to solution by 25.5 %. The discrepancy of $16.3\,\% = (25.5 - 9.2)\,\%$

originates from the reduced scalability of some subroutines of COSMO-CLM in coupled mode, which is probably related to sharing of storage space between COSMO-CLM and ECHAM if running on the same core in coupled mode. In particular the COSMO-CLM interface and the physics computations show almost no speed-up.

As demonstrated, the implementation of the usage of 3D-field exchange and of an alternating pro-

cesses distribution lead to an overall reduction of the total time to solution and costs of the coupled system CCLM+MPI-ESM by approximately 40 %. This corresponds to approximately 387 % of the $CCLM_{sa}$ costs.

### 4.5.2  Additional costs and time to solution

Several of the couplings investigated exhibit unnecessarily high costs of individual components

and/or a lack of scalability. This can originate from additional computations, from a different behaviour of the model components if coupled and/or from specific properties of the machine used.

The scalability results of all coupled components exhibit a weak scaling of parts of VEG3D, TRIMNP, CICE and ECHAM. In the CCLM+VEG3D coupling, this circumstance is negligible because the main costs lie with COSMO-CLM. However, all other component models make an efficient

coupling at higher speed rather difficult (see Fig. 5).

An analysis of the origin of increased time to solution and/or costs of the component models in coupled mode requires the availability of a model-internal analysis of timing. This information is available for the CCLM+MPI-ESM coupling.

Figures 6 and 7 show the time to solution and costs of the model system components, of the

CCLM+MPI-ESM coupled system and of the "improved" coupled system and its components. The latter are calculated by neglecting two of the additional computations, which, first, have been found to be responsible for the major part of the additional time to solution and, second, can be replaced by significantly more efficient alternative methods.



The first computation neglected is the calculation of horizontal derivatives executed in the ECHAM
component interface (see 3.2). It increases the costs from 170 HPSY (ECHAM (improved)) to
620 HPSY (ECHAM; see Fig. 7) if 32 cores are used for CCLM+MPI-ESM. This has two reasons:
First, a costly third-order spline method is used. This can be replaced by a fourth-order explicit
interpolation. Second, the calculation can be executed only on one core due to a lack of a halo in
ECHAM needed for the exchange of neighboring grid point values among cores with a common
boundary. This leads to a substantial load imbalance (not seen by LUCIA) and a fast loss of parallel
efficiency with increasing number of cores. To overcome this problem, there are two possibilities:
Either halos are introduced in ECHAM, which is planned for the upcoming ECHAM model version
or the derivatives are calculated in COSMO-CLM and sent to ECHAM additionally to the absolute
fields. The second option is the preferred one. ECHAM (improved) is the fastest and second-cheapest
(after MPIOM) of the coupled models.

The second additional computation neglected is the vertical interpolation of the exchanged model
variables in COSMO-CLM. It increases the costs from 310 HPSY (CCLM (improved)) to 430 HPSY
(CCLM; see Fig. 7). The interpolation method used is a spline interpolation, which is a rather costly
interpolation and which can be replaced by a second-to-fourth order explicit interpolation.

A neglection of the two inefficient additional computations decreases the costs from 1050 (CCLM+MPI-
ESM) to 480 (CCLM+MPI-ESM (improved)) CHPSY if 32 cores are used and from 3100 to 850 CHPSY
if 128 cores are used. It reduces the time to solution from 34.8 HPSY to 17 HPSY if 32 cores
are used and from 26 to 6.8 HPSY if 128 cores are used (see Fig. 6). Using 32 cores the costs of
CCLM+MPI-ESM (improved) are 108 % higher and the time to solution is 372 % longer. Using 128
cores, the costs are 234 % higher and the time to solution is 88 % longer than for $CCLM_{sa,sc}$. Thus,
CCLM+MPI-ESM (improved) can have a time to solution, which is comparable to $CCLM_{sa}$ and
other couplings at 30 % higher costs. However, this improvement of the computational performance
remains for future work.

## 5 Conclusions

We present couplings between the regional land-atmosphere climate model COSMO-CLM and
two land surface schemes (VEG3D, CLM), two ocean models (NEMO, TRIMNP+CICE) for the
Mediterranean Sea and for the North and Baltic Sea and the global atmosphere of MPI-ESM earth
system model using the fully parallelized coupler OASIS3-MCT. A unified OASIS3-MCT interface
(UOI) was developed and successfully applied for all couplings. All couplings are organized in a
least intrusive way such that the modifications of all model components are mainly limited to the
call of two subroutines receiving and sending the exchanged fields (as shown in Fig. 8 to 14). The
next step is development of the UOI for multiple couplings which allows regional climate system
modelling over Europe.



A series of simulations has been conducted with an aim to analyze the computational performance
of the couplings. The CORDEX-EU grid configuration of COSMO-CLM on a common computing
system (*Blizzard* at DKRZ) has been used in order to keep the results for time to solution, costs and
parallel efficiency comparable.

The results confirm that parallel efficiency is decreasing substantially if the number of grid points
per core is well below 100. For the configuration used (120x110 grid points), this limits the number
of nodes, which can be used efficiently to approximately four (128 cores or 256 threads).

The LUCIA tool of OASIS3-MCT has been used to measure the computing time used by each
model component and the coupler for communication and horizontal interpolation in dependence
on the computing resources used. This allows an estimation of the computing time for intermediate
computing resources and thus determination of an optimum configuration based on a limited number
of measurements. Furthermore, the scaling of each model component of the coupled system can be
analyzed and compared with that of the model in stand-alone mode. Thus, the additional costs of the
coupling and the origins of the relevant additional costs are measured.

The scaling of COSMO-CLM was found to be very similar in stand-alone and in coupled mode.
The weaker scaling, which occured in some configurations, was found to originate from additional
computations which do not scale but are necessary for coupling. In some cases the model physics or
the I/O routines exhibited a weaker scaling; most probably due to limited memory.

For the first time a sequential coupling of approximately 450 2D fields using the parallelized
coupler OASIS3-MCT was investigated. It was shown that the direct costs of coupling by OASIS3-
MCT (interpolation and communication) are negligible in comparison with the costs of the coupled
atmosphere-atmosphere model system. We showed that the exchange of one (pseudo-)3D field in-
stead of many 2D fields reduces the costs of communication drastically. Furthermore, the idling of
cores due to sequential coupling could be avoided by a dedicated launching of one process of each of
the two sequentially running models on each core making use of the multi-threading mode available
on *Blizzard*.

Inconsistencies of the time to solution of approximately 10 % were found between measurements
obtained from simulations conducted at two different physical times. This gives a measure of the
dependency of the time to solution on the status of the machine used, particularly originating from
the I/O.

A strategy for finding an optimum configuration was developed. Optimum configurations were
identified for all investigated couplings considering all three aspects of climate modeling perfor-
mance: time to solution, costs and parallel efficiency. The optimum configuration of coupled sys-
tems, that involve a component not scaling well with the available resources, is suggested to have
an acceptable cost considering the time to solution. This is the case for CCLM+MPI-ESM and
CCLM+TRIMNP+CICE couplings. An exception is the CCLM+VEG3D coupling. VEG3D was



found to have a weak scaling but a small work-load in comparison to COSMO-CLM. Thus, it has minimal impact on the performance of the coupled system.

The analysis of the optimum configurations led to the identification of a weak scalability of the MPI-ESM, CICE and VEG3D model components and high costs of additional computations in COSMO-CLM when coupled with MPI-ESM or CLM (see line 15, table 8). A detailed analy-

sis of the origin of weak scalability and/or increased costs was based on the time measurements of the subroutines of the model components which was only available for CCLM+MPI-ESM. The quantification of the additional costs at different configurations helped to analyze the potential of improved performance by replacing the non-parallel derivatives calculations and spline interpolation by parallel and explicit methods respectively. A direct comparison of the land model cou-

plings exhibits doubling costs in comparison with COSMO-CLM stand-alone and higher costs for CCLM+CLM than for CCLM+VEG3D due to higher costs of additional computations in COSMO-CLM. The direct comparison of the ocean couplings shows doubling costs for NEMO and increase by a factor of 2.5 for the CCLM+TRIMNP+CICE coupling. A direct comparison between NEMO and TRIMNP+CICE is not possible because the costs of NEMO-NORDIC have not been measured

on the same machine and for the same configuration. The lower parallel efficiency and costs of TRIMNP in comparison with NEMO-MED12 might result from the smaller number of grid points in the North and Baltic Sea than in the Mediterranean Sea.

The application of the procedure of finding an optimum configuration presented here is a useful step of development of a Regional Climate System Model coupling several model components. It

provides useful information on the bottle-necks of each coupling and helps in estimating the time to solution, costs and parallel efficiency of different couplings as a starting point for finding an optimum coupling layout and configuration for multiple couplings. It is applicable to each coupling layout and thus it could be as well very helpful for an efficient usage of other coupled model systems.

*Source code availability:* COSMO-CLM is an atmosphere model coupled to the soil-vegetation

model TERRA. Additionally, there is the possibility to use COSMO-ART (Vogel et al., 2009), which enables the calculation of transports of trace gases, aerosol and their interaction with atmospheric radiation and each other (atmospheric chemistry). Other regional processes in the climate system like ocean and ice sheet dynamics, plant responses, aerosol-cloud interaction, and the feedback to the GCM driving the RCM is made available by coupling COSMO-CLM via OASIS3-MCT with other

models. The development of fully coupled COSMO-CLM is an ongoing research project within the CLM-Community.

The COSMO-CLM model source code is freely available for scientific usage by members of the CLM-Community. The CLM-Community (www.clm-community.eu) is a network of scientists who accept the CLM-Community agreement. To become a member, please contact the CLM-Community

coordination office at DWD, Germany (clm-coordination@dwd.de).



The current released and evaluated climate model version of the CLM-Community is `COSMO_5.0_clm2`. It comes together with a recommendation for the configurations for the European domain.

**Appendix A: Model time step organisation**

In the following, the time step organisation within the coupled models is described. This aims at providing a basis of understanding of the coupling between the models.

**A1 COSMO-CLM**

Figure 8 gives an overview of the model initialization procedure, of the *Runge-Kutta* time step loop and of final calculations. The subroutines that contain all modifications of the model necessary for coupling are highlighted in red.

At the beginning ($t = t_m$) of the COSMO-CLM time step $(\Delta t)_c$ in `initialize_loop` the lateral, top and the ocean surface boundary conditions are updated. In `organize_data` the future boundary conditions at $t_f \geq t_m + \Delta t_c$ on the COSMO grid are read from a file (if necessary). As next `send_fld` and `receive_fld` routines are executed sending the COSMO-CLM fields to or receiving them from OASIS3-MCT in coupled simulations (if necessary). The details including the positioning of the `send_fld` routines will be explained in section 3.2 to 3.5.

At the end of the `initialize_loop` routine the model variables available at previous $t_p \leq t_m$ and next time $t_m < t_f$ of boundary update are interpolated linearly in time (if necessary) and used to initialize the boundlines of the COSMO-CLM model grid at the next model time level $t_m + (\Delta t)_c$ for the variables u and v wind, temperature and pressure deviation from a reference atmosphere profile, specific humidity, cloud liquid and ice water content, surface temperature over water surfaces and - in the boundlines only - surface specific humidity, snow surface temperature and surface snow amount.

In `organize_physics` all tendencies due to physical parameterizations between the current $t_m$ and the next time level $t_m + (\Delta t)_c$ are computed in dependence on the model variables at time $t_m$. Thus, they are not part of the Runge-Kutta time stepping. In `organize_dynamics` the terms of the Euler equation are computed.

The solution at the next time level $t_m + (\Delta t)_c$ is relaxed to the solution prescribed at the boundaries using an exponential function for the lateral boundary relaxation and a cosine function for the top boundary Rayleigh damping (Doms and Baldauf, 2015). At the lower boundary a slip boundary condition is used together with a boundary layer parameterisation scheme (Doms et al., 2011).



## A2 MPI-ESM

Figure 9 gives an overview of the ECHAM leapfrog time step (see DKRZ (1993) for details). Here the fields at time level $t_{n+1}$ are computed by updating the time level $t_{n-1}$ using tendencies computed at time level $t_n$.

After model initialization in `initialize` and `init_memory` and reading of initial conditions in `iorestart` or `ioinitial` the time step begins in `stepon` by reading the boundary conditions for the coupled models in `bc_list_read` if necessary, in this case for the ocean model MPIOM. In `couple_get_o2a` the fields sent by MPIOM to ECHAM (SSTs, SICs) for time level $t_n$ are received if necessary.

The time loop (`stepon`) has three main parts. It begins with the computations in spectral space, followed by grid space and spectral-space computations. In `scan1` the spatial derivatives (`sym2`, `ewd`, `fft1`) are computed for time level $t_n$ in Fourier space followed by the transformation into grid-space variables on the lon/lat grid. Now, the computations needed for two-way coupling with COSMO-CLM (`twc`) are done for time level $t_n$ variables followed by advection (`dyn`, `ldo_advection`)

at $t_n$, the second part of the time filtering of the variables at time $t_n$ (`tf2`), the calculation of the advection tendencies and update of fields for $t_{n+1}$ (`ldo_advection`). Now, the first part of the time filtering of the time level $t_{n+1}$ (`tf1`) is done followed by the computation of physical tendencies at $t_n$ (`physc`). The remaining spectral-space computations in `scan1` begin with the reverse fourier transformation (`fftd`).

## 1135   A3 NEMO-MED12

In Fig. 10 the flow diagram of NEMO 3.3 is shown. At the beginning the mpp communication is initialized by `cpl_prism_init`. This is followed by the general initialisation of the NEMO model. All OASIS3-MCT fields are defined inside the time loop, when `sbc` (surface boundary conditions) is called the first time. In `sbc_cpl_init` the variables which are sent and received are defined

over ocean and sea ice if applicable. At the end of `sbc_cpl_init` the grid is initialized, on which the fields are exchanged. In `cpl_prism_rcv` NEMO receives from OASIS3-MCT the fields necessary as initial and upper boundary conditions. NEMO-MED12 and NEMO-Nordic follow the time lag procedure of OASIS3-MCT appropriate for concurrent coupling. NEMO receives the restart files provided by OASIS3-MCT containing the COSMO-CLM fields at restart time. At all following cou-

pling times the fields received are not the COSMO-CLM fields at the coupling time but at a previous time, which is the coupling time minus a specified time lag. If a sea ice model is used, the fluxes from COSMO-CLM to NEMO have to be modified over surfaces containing sea ice. Hereafter, NEMO is integrated forward in time. At the end of the time loop in `sbc_cpl_snd` the surface boundary conditions are sent to COSMO-CLM. After the time loop integration the mpp communication is

finished in `cpl_prism_finalize`.



## A4 TRIMNP+CICE

Figures 11 and 12 show the flow diagrams of TRIMNP and CICE in which red parts are modifications of the models and blue parts are additional computations necessary for coupling. First, initialization is done by calling `init_mpp` and `cice_init` in TRIMNP and CICE, respectively.

In `cice_init`, the model configuration and the initial values of variables are set up for CICE while for TRIMNP `setup_cluster` is used for the same purpose. In both models the receiving (`ocn_receive_fld`, `ice_receive_fld`) and sending (`ocn_send_fld`, `ice_send_fld`) subroutines are used in the first time step ($t = 0$) prior to the time loop to provide the initial forcing. The time loop of TRIMNP covers a grid loop in which several grids on higher resolutions are

potentially *one-way* nested for specific sub-regions with rather complex bathymetry, e. g. Kattegat of the North Sea. Note that for the coupling, only the first/main grid is applied. The grid loop begins with `rcv_parent_data` that sends data from the coarser grid to the nested grid. Then, `do_update` updates the forcing data passed from COSMO-CLM and CICE as well as the lateral boundary data are read from files. After updating, the physics and dynamics computations are

mainly done in `heat_flux`, `turbo_adv`, `turbo_gotm`, `do_constituent`, `do_explicit` and `do_implicit`. At the end of the grid loop, the main grid sends data to the finer grid by calling `snd_parent_data` if necessary. At the end of each time step, output and restart data are written to files. Eventually, `stop_mpp` is called at the end of the main program to de-allocate the memory of all variables and finalize the program.

The time loop of CICE has two main parts. In the first part `ice_step`, physical, dynamical and thermo-dynamical processes of the time step $t = t_n$ are mainly computed in `step_therm1`, `step_therm2`, `step_radiation`, `biogeochemistry` and `step_dynamics`, followed by `write_restart` and `final_restart` for writing the output and restart files. Then, the time step is increased to a new time step $t = t_{n+1}$, followed by an update of forcing data from COSMO-

CLM and TRIMNP via `ice_receive_fld` if necessary and a sending of fields to COSMO-CLM and TRIMNP via `ice_send_fld`. At the end of the time loop, all file units are released in `release_all_fileunits` and `oas_ice_finalize` concludes the main program.

## A5 VEG3D

Figure 13 shows the flow diagramm of VEG3D for the coupled system. In a first step the subroutine

`oas_veg3d_init` is called in order to initialize the MPI communication for the coupling. Afterwards, the model setup is specified by reading the VEG3D namelist and by loading external landuse and soil datasets. The definition of the grid and the coupling fields is done in `oas_veg3d_define`. The main program includes two time loops. In the first time loop vegetation parameters are calculated for every simulated day. In the second loop (over the model time steps) the coupling fields

from COSMO-CLM are received via OASIS3-MCT in `receive_fld_2cos` at every coupling

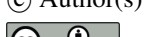



time step. Using these updated fields the energy balance of the canopy for the current time level $t_n$ is solved iterativly and based on this the latent and sensible heat fluxes are calculated. The heat conduction and the Richardson equation for the time level $t_{n+1}$ are solved by a semi-implicit Crank-Nicholson method. After these calculations the simulated coupling fields from VEG3D are sent to

COSMO-CLM in `send_fld_2cos`. At the end, output and restart files are written for selected time steps. The `oas_veg3d_finalize` subroutine stops the coupling via OASIS3-MCT.

### A6   CLM

CLM is embedded within the CESM modelling system and its multiple components. In the case of land-only simulations, the active components are the driver/internal coupler (CPL7), CLM and a data

atmosphere component. The later is substituted to the atmospheric component used in coupled mode and provides the atmospheric forcing usually read from a file. In the framework of the OASIS3-MCT coupling, however, the file reading is deactivated and replaced by the coupling fields received from OASIS3-MCT (`receive_field_2cos`). The send operation (`send_field_2cos`) is also positioned in the data atmosphere component in order to enforce the same sequence of calls as in

CESM. The definition of coupling fields and grids for the OASIS3-MCT coupling is also done in the data atmosphere component during initialization before the time loop. Additionally, the initialization (`oas_clm_init`) and finalization (`oas_clm_finalize`) of the MPI communicator for the OASIS3-MCT coupling is positioned in the CESM driver, respectively before and after the time loop. The sequence of hydrological and biogeophysical calculations during the time loop are given

in black and the calls to optional modules are marked in grey.

*Acknowledgements.* The development of COSMO-CLM couplings would have not been possible without the continuous work done by OASIS, COSMO and CLM-Community colleagues and provision of computing time and support by computing centers. In particular we would like to thank Ulrich Schaettler (DWD) and Hans-Jürgen Panitz (KIT Karlsruhe) for source code maintenance of COSMO and COSMO-CLM. The overall support

and provision of computing time by DKRZ Hamburg and the hosting of a developers workshop by CSCS Lugano are highly acknowledged.

     Finally, we would like to highlight the contributions to the work presented here by furhter colleagues. First of all, Irina Fast provided the solution for dedicated distribution of model tasks on cores and the MPI-ESM version using the OASIS3-MCT coupler. Andreas Dobler (FU Berlin) made the pioneering work in coupling

of COSMO-CLM using OASIS3. Sophie Valcke (CERFACS) provided the OASIS3-MCT support necessary to solve the problems of coupling with a regional model. Last but not least, Matthieu Leclair (ETH Zürich) helped to improve the manuscript a lot.





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



Table 1: **List of acronyms** used throughout the paper

| Acronym | Meaning |
| --- | --- |
| COSMO | Limited-area model of the COnsortium for Small-scale MOdelling |
| COSMO-CLM | COSMO model in CLimate Mode |
| CCLM | Short for COSMO-CLM used in figures, tables, formulas and coupled-system acronyms |
| CLM | Community Land Model of NCAR |
| VEG3D | Soil and vegetation model of KIT |
| NEMO | Community model 'Nucleus for European Modelling of the Ocean' |
| TRIMNP | Tidal, Residual, Intertidal mudflat Model Nested parallel Processing regional ocean model |
| CICE | Sea ice model of LANL |
| MPI-ESM | Global Earth System Model of MPIfM Hamburg |
| ECHAM | Atmosphere model (ECMWF dynamics and MPIfM Hamburg physics) of MPI-ESM |
| MPIOM | MPIfM Hamburg Ocean Model of MPI-ESM |
| OASIS3-MCT | Coupling software for Earth System Models of CERFACS |
| CESM | Community Earth System Model |
| Institutions | |
| MPIfM | Max-Planck-Institut für Meteorologie Hamburg, Germany |
| LANL | Los Alamos National Laboratory, USA |
| CERFACS | Centre Europeen de Recherche et de Formation Avancee en Calcul Scientifique, Toulouse, France |
| CLM-Community | Climate Limited-area Modelling (CLM-)Community |
| ECMWF | European Center for Medium Range Weather Forecast, Reading, Great Britain |
| NCAR | National Center for Atmospheric Research, Boulder, USA |
| CNRS | Centre National de Recherche Scientifique, Paris, France |
| ETH | Eidgenössische Technische Hochschule, Zürich, Switzerland |
| KIT | Karlsruher Institut für Technologie, Germany |
| GUF | Goethe-Universität Frankfurt am Main, Germany |
| HZG | Helmholtz-Zentrum Geesthacht, Germany |
| BTU | Brandenburgische Technische Universität Cottbus-Senftenberg, Cottbus, Germany |
| FUB | Freie Universität Berlin, Germany |
| Model domains | |
| CORDEX-EU | CORDEX domain for regional climate simulations over Europe |





Table 2: **Coupled model systems**, their components and the institution at which they are used. For the meaning of acronyms see Table 1.

| Coupled model system | Institution | First coupled component | Second coupled component |
| --- | --- | --- | --- |
| CCLM+CLM | ETH | CLM | – |
| CCLM+VEG3D | KIT | VEG3D | – |
| CCLM+NEMO-MED12 | GUF | NEMO-MED12 | – |
| CCLM+TRIMNP+CICE | HZG | TRIMNP | CICE |
| CCLM+MPI-ESM | BTU and FUB | ECHAM | MPIOM |





Table 3: **Properties of the coupled model components.** For the meaning of acronyms see Table 1. The configuration used is a coarse-grid regional climate simulation configuration used for sensitivity studies, tests and continental-scale climate simulations.

| model | CCLM | CLM | VEG3D | MPI-ESM |
|---|---|---|---|---|
| Full name | COSMO model in climate mode | Community Land Model | Vegetation model | Max Planck Institute Earth System Model |
| Institution | CLM-Community | NCAR and other institutions | KIT | MPIfM Hamburg |
| Coupling area | CORDEX-EU | CORDEX-EU land | CORDEX-EU land | CORDEX-EU |
| Horizontal res. (km) | 50 | 50 | 50 | 330 |
| Nr. of levels | 40/45 | 15 | 10 | 47 |
| Time step (s) | 300 | 300 | 300 | 600 |
| Reference | (Baldauf et al., 2011) | (Oleson et al., 2010) | (Schädler, 1990) | (Stevens et al., 2013) |

| model | NEMO-MED12 | NEMO-NORDIC | TRIMNP | CICE |
|---|---|---|---|---|
| Full name | Nucleus for European Modelling of the Ocean - Mediterranean Sea | Nucleus for European Modelling of the Ocean - North and Baltic Sea | Tidal, Residual, Intertidal mudflat Model Nested parallel Processing | Sea Ice Model |
| Institution | CNRS | CNRS | Univ. Trento, HZG | LANL |
| Coupling area | Mediterranean Sea (without Black Sea) | North and Baltic Sea | North and Baltic Sea | Baltic Sea and Kattegat |
| Horizontal res. (km) | 6-8 | 3.7 | 12.8 | 12.8 |
| Nr. of levels | 50 | 56 | 50 | 5 |
| Time step (s) | 720 | 300 | 240 | 240 |
| Reference | Madec (2008); Lebeaupin et al. (2011); Akhtar et al. (2014) | Hordoir et al. (2013), Dieterich et al. (2013); Pham et al. (2014) | Casulli and Cattani (1994), Casulli and Stelling (1998); Ho-Hagemann et al. (2013) | Hunke et al. (2013); Ho-Hagemann et al. (2013) |



Table 4: **Variables exchanged between CCLM and the global model MPI-ESM.** The CF standard-names convention is used. Units are given as defined in CCLM. $\otimes$: information is sent by CCLM; $\odot$: information is received by CCLM. *3D* indicates that a 3-dim. field is sent/received.

| Variable (unit) | CCLM+MPI-ESM |
|---|---|
| Temperature $(K)$ | $\odot\otimes 3D$ |
| U-component of wind $(m\,s^{-1})$ | $\odot\otimes 3D$ |
| V-component of wind $(m\,s^{-1})$ | $\odot\otimes 3D$ |
| Specific humidity $(kg\,kg^{-1})$ | $\odot\otimes 3D$ |
| Specific cloud liquid water content $(kg\,kg^{-1})$ | $\odot\otimes 3D$ |
| Specific cloud ice content $(kg\,kg^{-1})$ | $\odot\otimes 3D$ |
| Surface pressure $(Pa)$ | $\odot\otimes$ |
| Sea surface temperature $SST$ $(K)$ | $\odot$ |
| Surface snow amount $(m)$ | $\odot$ |
| Surface geopotential $(m\,s^{-2})$ | $\odot$ |

$SST = (sea\_ice\_area\_fraction \cdot T_{sea\,ice}) + (SST \cdot (1 - sea\_ice\_area\_fraction))$





Table 5: As Table 4 but **variables exchanged between CCLM and the ocean models NEMO, TRIMNP and CICE.**

| Variable (unit) | CCLM+ NEMO-MED12 | CCLM+ NEMO-NORDIC | CCLM+ TRIMNP+ CICE |
|---|:---:|:---:|:---:|
| Sea surface temperature ($K$) | ⊙ | ⊙ | ⊙ |
| 2 m temperature ($K$) | – | – | ⊗ |
| Potential temperature NSL ($K$) | – | – | ⊗ |
| Temperature NSL ($K$) | – | – | ⊗ |
| Sea ice area fraction (1) | – | ⊙ | – |
| Surface pressure ($Pa$) | – | ⊗ | – |
| Mean sea level pressure ($Pa$) | – | – | ⊗ |
| Surface downward east- and northward stress ($Pa$) | ⊗ | ⊗ | – |
| Surface net downward shortwave flux ($W\,m^{-2}$) | ⊗ | ⊗ | ⊗ |
| Surface net downward longwave flux ($W\,m^{-2}$) | – | – | ⊗ |
| Non-solar radiation $NSR$ ($W\,m^{-2}$) | ⊗ | ⊗ | – |
| Surface downward latent heat flux ($W\,m^{-2}$) | – | – | ⊗ |
| Surface downward heat flux $HFL$ ($W\,m^{-2}$) | – | – | ⊗ |
| Evaporation-Precipitation $E-P$ ($kg\,m^{-2}$) | ⊗ | ⊗ | – |
| Total precipitation flux $TPF$ ($kg\,m^{-2}\,s^{-1}$) | – | – | ⊗ |
| Rain flux $RF$ ($kg\,m^{-2}\,s^{-1}$) | – | – | ⊗ |
| Snow flux $SF$ ($kg\,m^{-2}\,s^{-1}$) | – | – | ⊗ |
| U- and V-component of 10 m wind ($m\,s^{-1}$) | – | – | ⊗ |
| 2 m relative humidity (%) | – | – | ⊗ |
| Specific humidity NSL ($kg\,kg^{-1}$) | – | – | ⊗ |
| Total cloud cover (1) | – | – | ⊗ |
| Half height of lowest CCLM level ($m$) | – | – | ⊗ |
| Air density NSL ($kg\,m^{-3}$) | – | – | ⊗ |

NSL = the lowest (near-surface) level of the 3-dimensional variable

NSR = surface net downward longwave flux + surface downward latent and sensible heat flux

HFL = surface net downward shortwave flux + surface downward longwave flux + surface downward latent and sensible heat flux

TPF = RF + SF = convective and large-scale rainfall flux + convective and large-scale snowfall flux

E-P = -(surface downward sensible heat flux / LHV) - TPF; LHV: Latent heat of vaporization = 2.501E6 J/kg



Table 6: As Table 4 but **variables exchanged between CCLM and the land surface models VEG3D and CLM.**

| Variable (unit) | CCLM+VEG3D | CCLM+CLM |
|---|---|---|
| Leaf area index (1) | ⊗ | – |
| Plant cover (1) | ⊗ | – |
| Vegetation function (1) | ⊗ | – |
| Surface albedo (1) | ⊙ | ⊙ |
| Height of lowest level ($m$) | – | ⊗ |
| Surface pressure ($Pa$) | ⊗ | – |
| Pressure NSL ($Pa$) | ⊗ | ⊗ |
| Snow flux $SF$ ($kg\,m^{-2}\,s^{-1}$) | ⊗ | ⊗ |
| Rain flux $RF$ ($kg\,m^{-2}\,s^{-1}$) | ⊗ | ⊗ |
| Temperature NSL ($K$) | ⊗ | ⊗ |
| Grid-mean surface temperature ($K$) | ⊙ | ⊙ |
| Soil surface temperature ($K$) | ⊙ | – |
| Snow surface temperature ($K$) | ⊙ | – |
| Surface snow amount ($m$) | ⊙ | – |
| Density of snow ($kg\,m^{-3}$) | ⊙ | – |
| Thickness of snow ($m$) | ⊙ | – |
| Canopy water amount ($m$) | ⊙ | – |
| Specific humidity NSL ($kg\,kg^{-1}$) | ⊗ | ⊗ |
| Surface specific humidity ($kg\,kg^{-1}$) | ⊙ | – |
| Subsurface runoff ($kg\,m^{-2}$) | ⊙ | – |
| Surface runoff ($kg\,m^{-2}$) | ⊙ | – |
| Wind speed $|\vec{v}|$ NSL ($m\,s^{-1}$) | ⊗ | – |
| U- and V-component of wind NSL ($m\,s^{-1}$) | – | ⊗ |
| Surface downward sensible heat flux ($W\,m^{-2}$) | ⊙ | ⊙ |
| Surface downward latent heat flux ($W\,m^{-2}$) | – | ⊙ |
| Surface direct and diffuse downwelling shortwave flux in air ($W\,m^{-2}$) | ⊗ | ⊗ |
| Surface net downward longwave flux ($W\,m^{-2}$) | ⊗ | ⊗ |
| Surface flux of water vapour ($s^{-1}\,m^{-2}$) | ⊙ | – |
| Surface downward east- and northward flux (U-/V-momentum flux, $Pa$) | – | ⊙ |

NSL = the lowest (near-surface) level of the 3-dimensional variable

RF = convective and large-scale rainfall flux; SF = convective and large-scale snowfall flux

SWD_S = surface diffuse and direct downwelling shortwave flux in air




Table 7: **Measures of computational performance** used for computational performance analysis.

| Measure (unit) | Acronym | Description |
|---|---|---|
| simulated years (1) | sy | Number of simulated physical years |
| number of cores (1) | n | Number of computational cores used in a simulation per model component |
| number of threads (1) | R | Number of parallel processes or threads configured in a simulation per model component. On *Blizzard* at DKRZ one or two threads can be started on one core. |
| time to solution $(HPSY)$ | T | Simulation time of a model component measured by LUCIA per simulated year |
| speed $(HPSY^{-1})$ | s | $= T^{-1}$ is the number of simulated years per simulated hour by a model component |
| costs $(CHPSY)$ | – | $= T \cdot n$ is the core hours used by a model component running on $n$ cores per simulated year |
| speed-up (%) | SU | $= \frac{HPSY_1(R_1)}{HPSY_2(R_2)} \cdot 100$ is the ratio of time to solution of a model component configured for reference and actual number of threads |
| parallel efficiency (%) | PE | $= \frac{CHPSY_1}{CHPSY_2} \cdot 100$ is the ratio of core hours per simulated year for reference $(CHPSY_1)$ and actual $(CHPSY_2)$ number of cores |

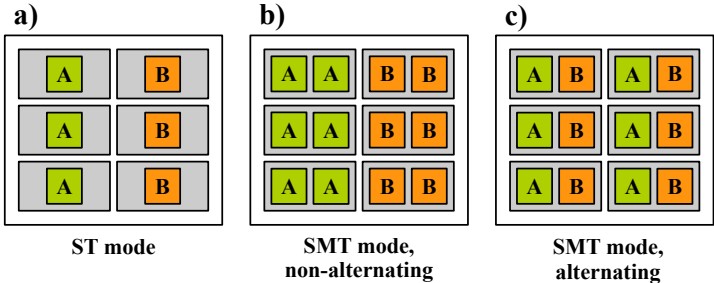

Figure 1: **Schematic processes distribution on a hypothetical computing node** with six cores (gray-shaded areas) in a) ST mode, b) SMT mode with non-alternating processes distribution and c) SMT mode with alternating processes distribution. "A" and "B" are processes belonging to two different parallel applications sharing the same node. In b) and c) two processes of the same (b) or different (c) application share one core using the simultaneous multi-threading (SMT) technique while in a) only one process per core is launched in the single-threading (ST) mode.





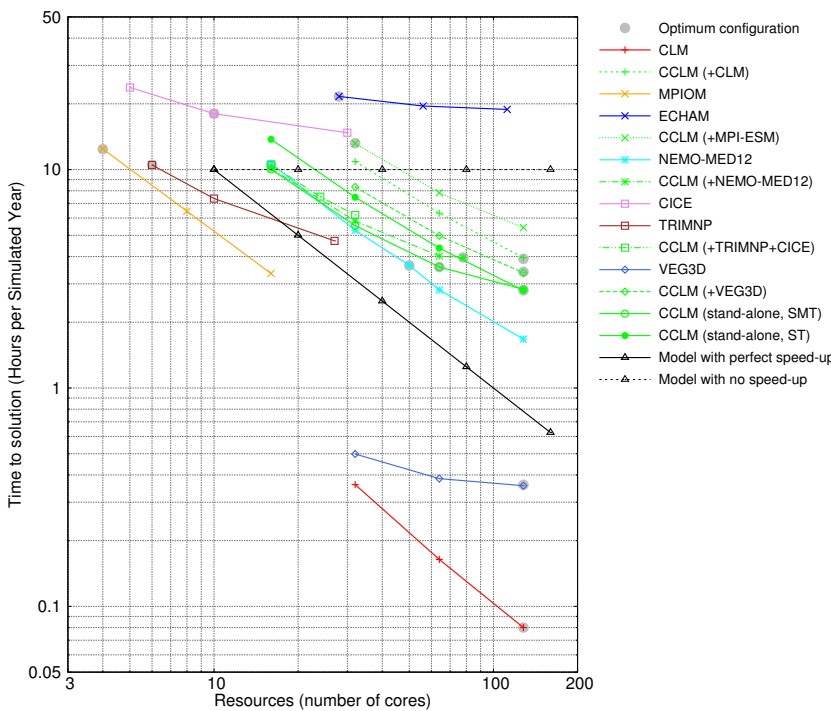

Figure 2: **Time to solution of model components** of the coupled systems (indicated for CCLM in brackets) and for CCLM stand-alone ($CCLM_{sa}$) in hours per simulated year (HPSY) in dependence on the computational resources (number of cores) in single threading (ST) and in multi threading (SMT) mode. The times for model components ECHAM and MPIOM of MPI-ESM are given separately. The optimum configuration of each component is highlighted by a gray dot. The hypothetical result for a model with perfect and no speed-up is given as well.





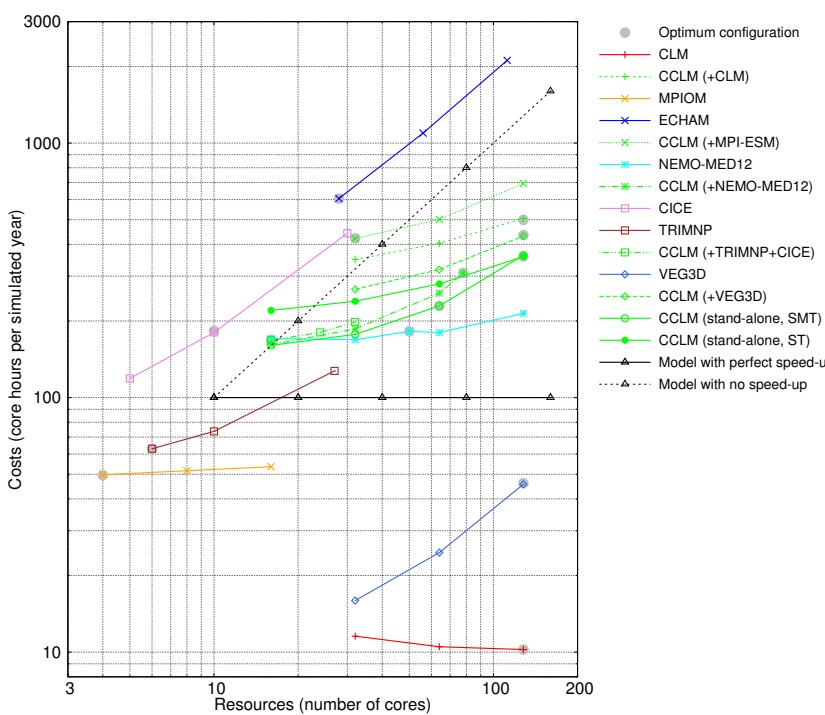

Figure 3: As Fig. 2 but for the **costs of the model components** in core hours per simulated year.





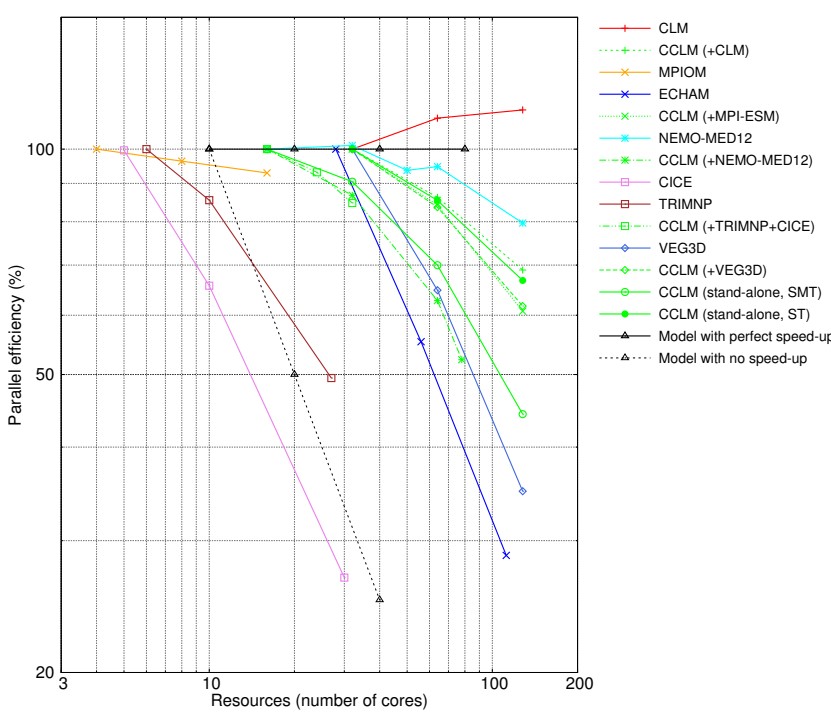

Figure 4: As Fig. 2 but for the **parallel efficiency of the model components** in % of the reference configuration.





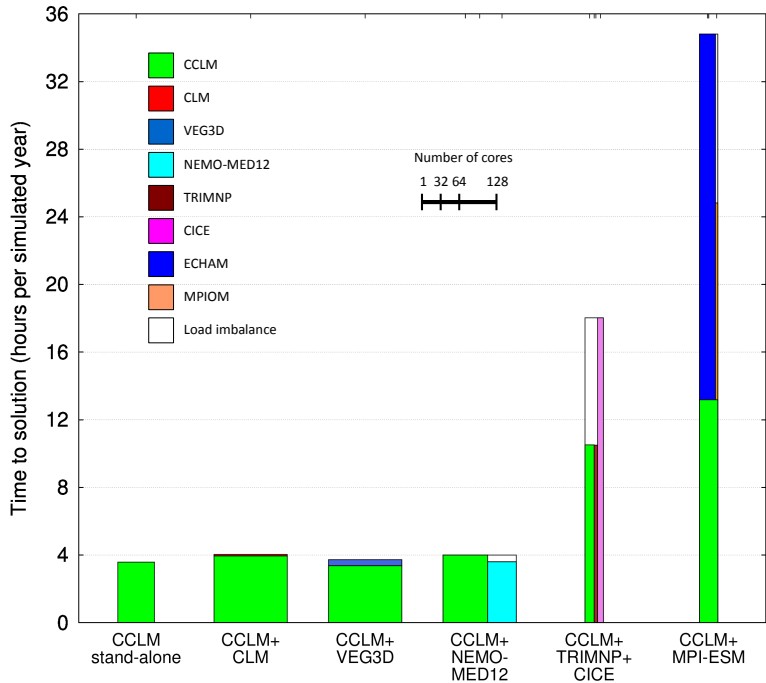

Figure 5: **Time to solution and costs of model components at optimum configuration** of cou-
plings investigated and of stand-alone CCLM. The boxes' widths correspond to the number of cores
used per component. The area of each box is equal to the costs (the amount of core hours per sim-
ulated year) consumed by each component. The white areas indicate the load imbalance between
concurrently running components. See Table 8 for details.





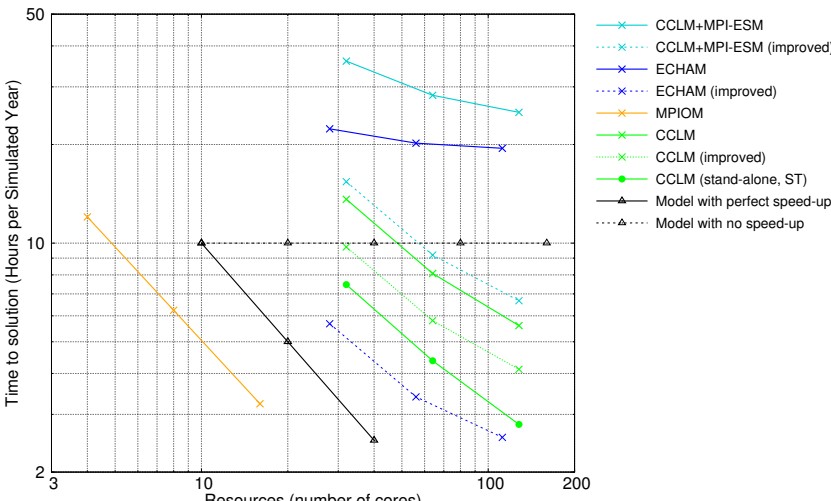

Figure 6: **As Fig. 2 but for CCLM+MPI-ESM coupling and its model components** for the reference and the improved version of the coupling. *Improved* coupling excludes in ECHAM the time for horizontal derivative calculation and in CCLM the time for the vertical interpolation (see section 4.5.2 for details).

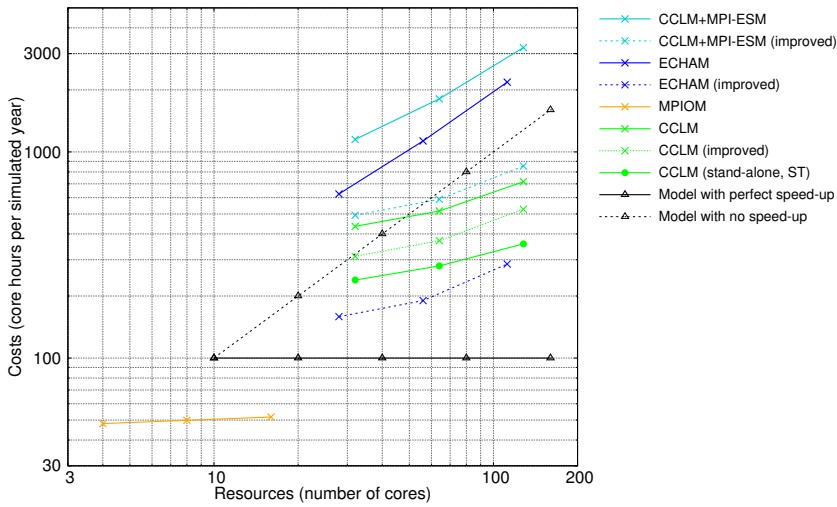

Figure 7: **As Fig. 6 but for costs.**





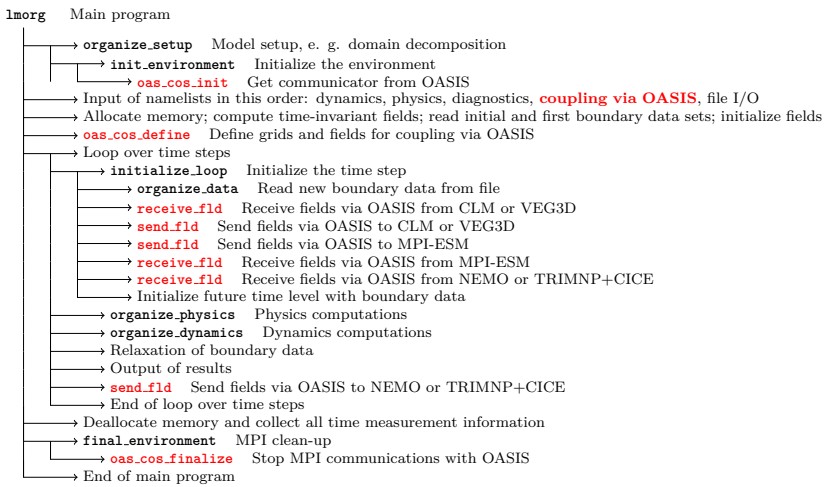

Figure 8: **Simplified flow diagram of the main program of the regional climate model COSMO-CLM**, version `4.8_clm19_uoi`. The red highlighted parts indicate the locations at which the additional computations necessary for coupling are executed and the calls to the OASIS interface take place. Where applicable, the component models to which the respective calls apply are given.




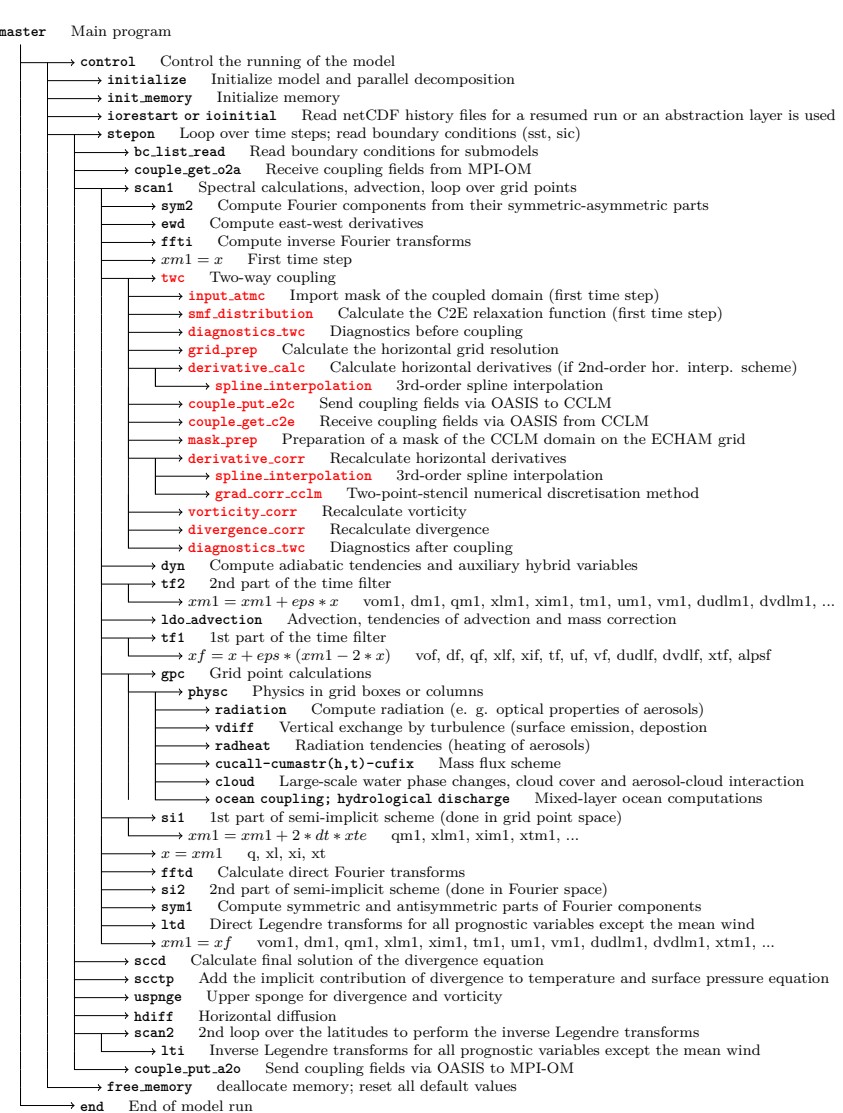

Figure 9: **As Fig. 8 but for the global atmosphere model ECHAM of MPI-ESM.**





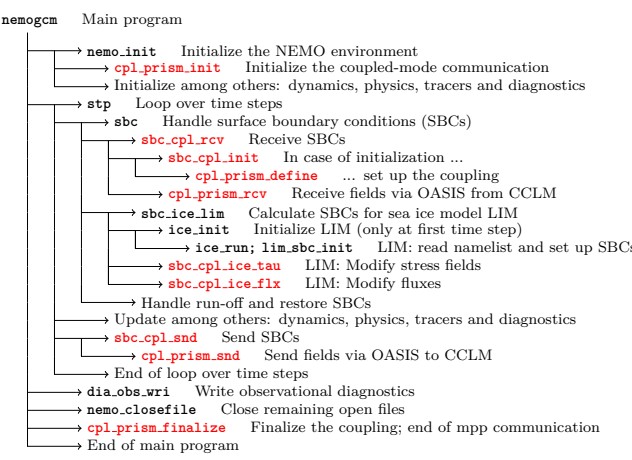

Figure 10: **As Fig. 9 but for the ocean model NEMO version 3.3.**





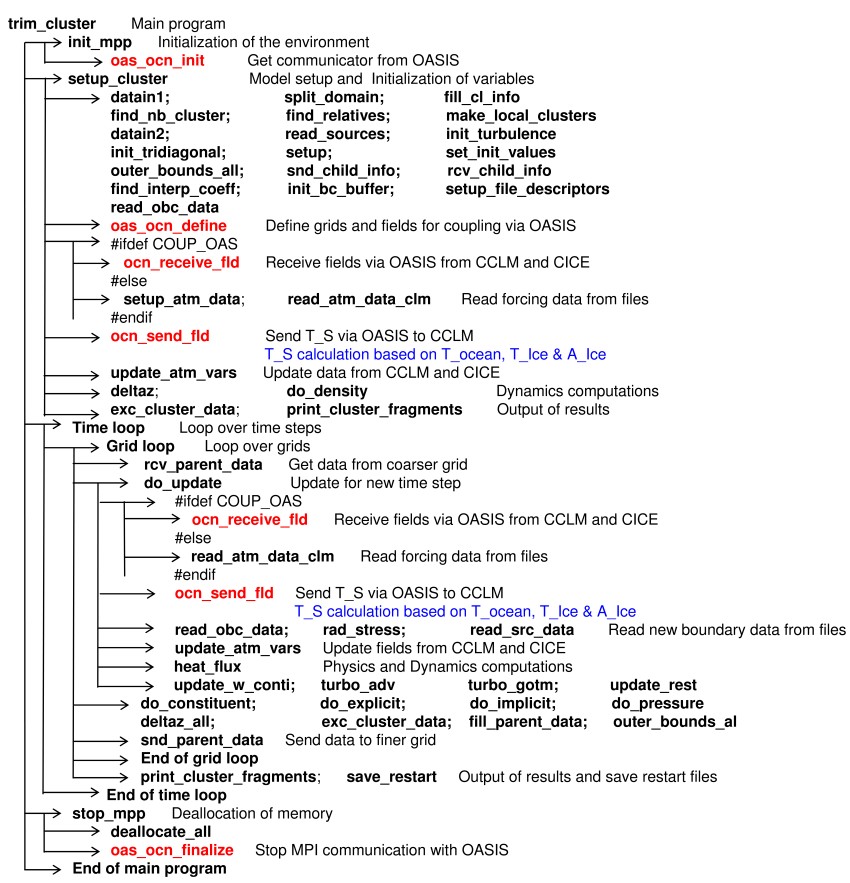

Figure 11: **As Fig. 9 but for the ocean model TRIMNP.**



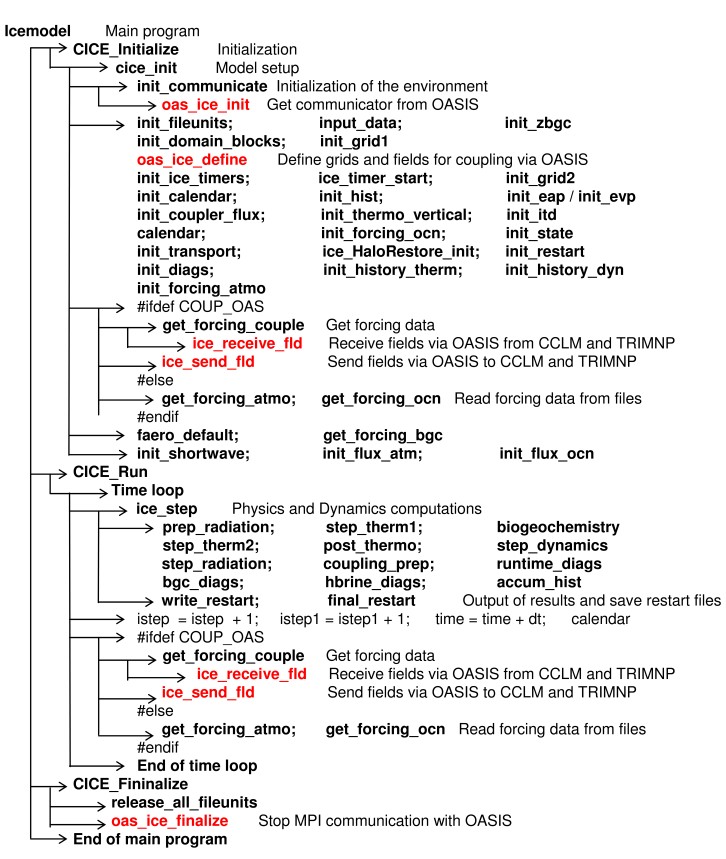

Figure 12: **As Fig. 9 but for the sea ice model CICE.**





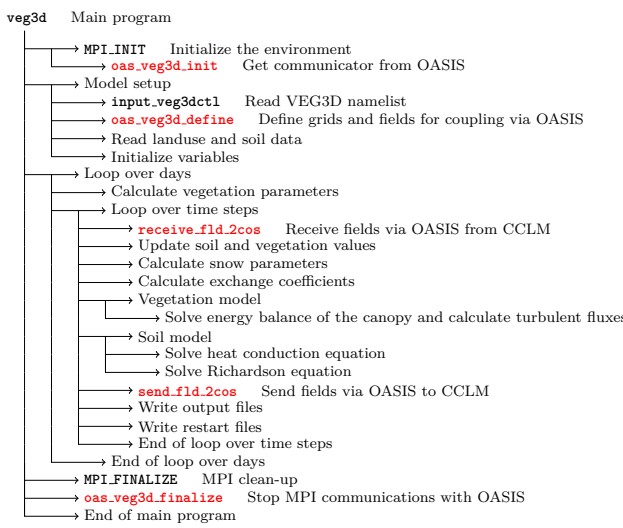

Figure 13: **As Fig. 9 but for the soil-vegetation model VEG3D.**




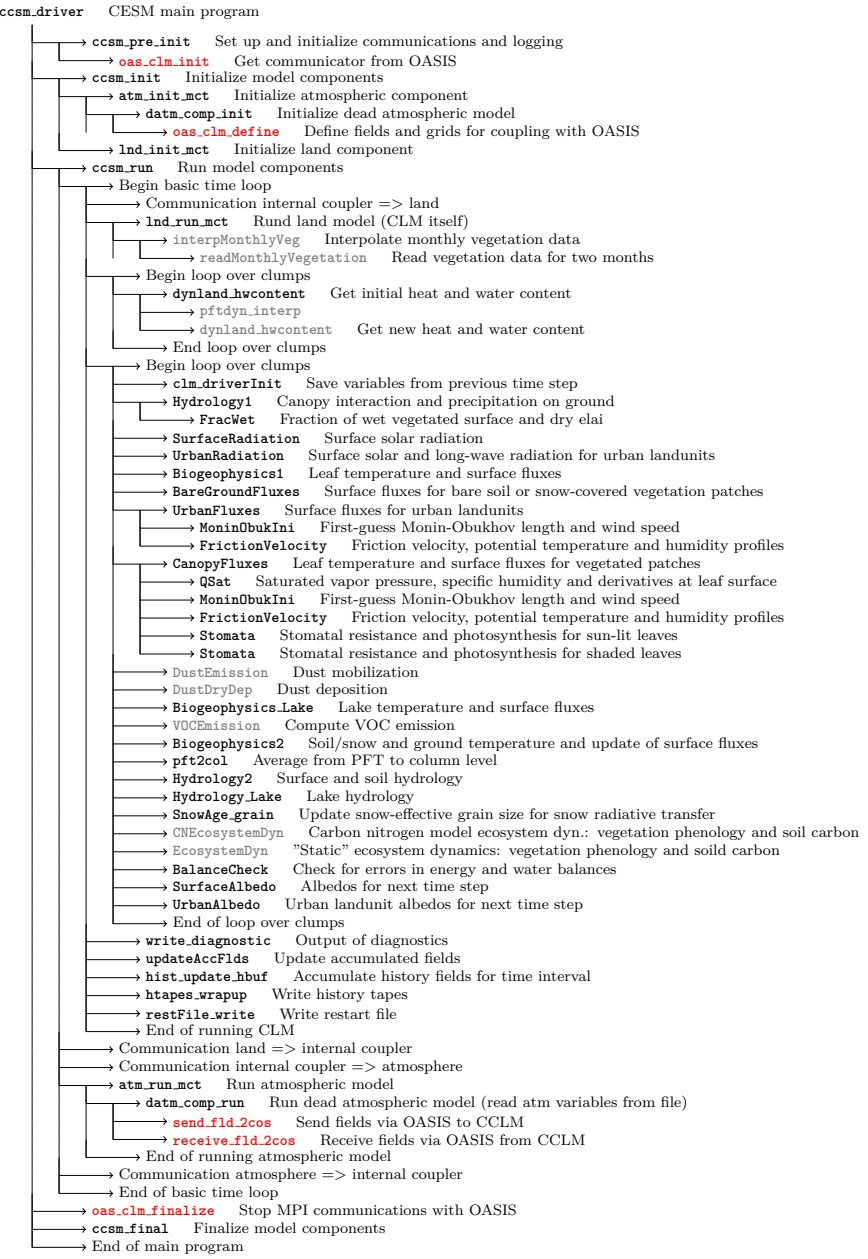

Figure 14: **As Fig. 9 but for the Community Land Model (CLM).** The gray highlighted routines
are optional.



Table 8: Analysis of the **optimum configurations of the coupled systems** (CS) given in the table header (compare to Fig. 5). *seq* refers to sequential and *con* to concurrent couplings. *Thread mode* is either the ST or the SMT mode (see Fig. 1). *APD* indicates whether an alternating processes distribution was used or not. *Time to solution (%)* and *Cost (%)* are caculated with respect to the reference, which is the CCLM stand-alone configuration $CCLM_{sa}$. The time to solution does not include the time needed for OASIS interpolations. $CS - CCLM_{sa}$ gives the differences between CS and the optimum $CCLM_{sa}$ configuration (in percent). *Oasis hor. interp.* and *load imbalance* are in percent of the costs of each CS. $CCLM - CCLM_{sa}$ gives the additional costs (in percent) of CCLM within a CS compared to the reference. $CCLM - CCLM_{sa,sc}$ gives the additional costst (in percent) of CCLM within a CS compared to CCLM stand-alone that used the same configuration.

| | | CCLM stand-alone | CCLM+ CLM | CCLM+ VEG3D | CCLM+ NEMO-MED12 | CCLM+ TRIMNP+ CICE | CCLM+ MPI-ESM |
|---|---|---|---|---|---|---|---|
| 1 | Type of coupling | – | seq | seq | con | con | seq and con |
| 2 | Thread mode | SMT | SMT | SMT | SMT | SMT | SMT |
| 3 | APD used | – | yes | yes | no | no | yes |
| 4 | # nodes | 2 | 4 | 4 | 4 | 1 | 1 |
| 5 | # cores per component | 64 | 128, 128 | 128, 128 | 78, 50 | 16, 6, 10 | 32, 28, 4 |
| 6 | Time to solution ($HPSY$) | 3.6 | 4.0 | 3.7 | 4.0 | 18.0 | 34.8 |
| 7 | Time to solution (%) | 100.0 | 111.1 | 102.8 | 111.1 | 450.0 | 866.7 |
| 8 | $CS - CCLM_{sa}$ | – | 11.1 | 2.8 | 11.1 | 350.0 | 766.7 |
| 9 | Costs ($CHPSY$) | 230.4 | 512.0 | 473.6 | 512.0 | 576.0 | 1113.6 |
| 10 | Costs (%) | 100.0 | 222.2 | 205.6 | 222.2 | 250.0 | 483.3 |
| 11 | OASIS hor. interp. | – | 3.0 | 0.0 | 0.03 | 0.3 | 0.7 |
| 12 | Load imbalance | – | – | – | 3.9 | 20.9 | 3.6 |
| 13 | $CS - CCLM_{sa}$ | – | 122.2 | 105.6 | 122.2 | 150.0 | 383.3 |
| 14 | $CCLM - CCLM_{sa}$ | – | 119.2 | 87.2 | 35.4 | -27.0 | 83.1 |
| 15 | $CCLM - CCLM_{sa,sc}$ | – | 40.9 | 20.4 | 17.2 | 4.9 | 76.4 |