# Peer review of "The regional climate model COSMO-CLM 4.8 coupled to regional ocean, land surface and global earth system models using OASIS3-MCT: description and performance"

_Geoscientific Model Development, 2016_

## Short Comment (SC1) · 20 Apr 2016

Dear authors,

In my role as Executive editor of GMD, I would like to bring to your attention our Editorial version 1.1:

http://www.geosci-model-dev.net/8/3487/2015/gmd-8-3487-2015.html

This highlights some requirements of papers published in GMD, which is also available on the GMD website in the 'Manuscript Types' section:

[Figure]

http://www.geoscientific-model-development.net/submission/manuscript_types.html

In particular, please note that for your paper, the following requirement has not been met in the Discussions paper:

- "The main paper must give the model name and version number (or other unique identifier) in the title."

Please add the version number of COSMO-CLM in the title upon your revised submission to GMD.

Additionally, I ask you to revise the Code Availability Section.
First of all it should be clearly maked as an individual section. But the type-setting of copernicus will ensure that anyway, but, secondly, the content of the Code Availability section is somewhat confusing: On the one hand side code parts not discussed in the article (COSMO-ART) are named. If it is not used in your article it should not be mentioned here. On the other hand side you write about a lot of other climate system models which are coupled to COSMO-CLM via OASIS3-MCT. Therefore the availability of these models should also be clarified in the Code Availability section. Last but not least, a kind reader would be interested how to access the COSMO-CLM version including OASIS3-MCT. For the latter it would be enough to say that it is available by contacting one of the authors and will be part of a future official COSMO(-CLM) version.

Yours,

Astrid Kerkweg

---

## Referee Comment (RC1) · Anonymous Referee #1 · 31 May 2016

Introduction

Coding and technical aspects of coupling Earth System Models are often relegated to institutional reports seldom referenced or widely read, and outcomes of work in coupling and load balancing are often blindly used by physical and biogeochemical modeling groups. Therefore I commend the authors for documenting their expansive coupling work, and for submitting it to be reviewed for a journal with a readership that bridges the coupler development and physical modeling communities. I do, however, have reservations about the final results, methods, and one comment about the scope

of the cited literature.

Significance of this paper in the context of other work

The manuscript's introduction could extend the perceived reach of the work if it were to illustrate its international significance. There are several latest-generation regional coupled earth system models in development in the U.S. and Canada, some of which use MCT and takes advantage of the work of Craig et al. (2012) that could have been cited and have appeared in the reviewed literature in recent years. The reason I mention these publications is to say that the introductory argument perhaps could be further enhanced, since work on load balancing high-resolution regional coupled earth system models is taking place in many parts of the Earth System Modeling community. This helps to widen the appeal of the current manuscript, and its significance.

Efficiency versus accuracy

This paper discusses a considerable number (five) of different coupled model configurations using CCLM, however only scant information is provided on each one of these configurations. It would be particularly useful to view maps of model domains to demonstrate the individual configurations for each of the coupled model systems in Table 2. This would help make it clear exactly how much ocean, land and sea ice exist in the respective model domains. Such details can have a large impact scalability and parallel efficiency, especially in the cryosphere (sea ice and snow). Therefore I suggest providing greater detail on the physical configuration of each of the models chosen, because this, too, has an enormous impact on the model solution. To illustrate this point, I focus here on the implementation of CICE Version 5 for CCLM+TRIMNP+CICE.

The computational efficiency of the solution in CICE is heavily dependent upon the total number of sea ice thickness categories used, the number of tracers needed, for example, by melt pond and ice-age tracking and biogeochemistry, and most importantly, the sea ice mechanics solution. If CICE 5 has been configured to use anisotropic (Elastic Anisotropic Plastic; EAP) sea ice mechanics, then it will definitely be expensive, and,

could take as much as 30% of the total model execution time in pan-arctic fully coupled regional models, if a highly converged plastic sea-ice solution is required (2-second sub-cycling). However, if using the Elastic Viscous Plastic (EVP) sea ice rheology with 10-second sub-cycling, the time to solution of the sea ice model greatly improves, with only slight degradation of the plastic solution. In this configuration, the sea ice model could take only 10% of the total core time of running the model. It is still unknown as to which of the two variants is physically more accurate. This is precisely the same CICE Version 5.1 code, in the same coupled framework, using MCT, but with two different namelist settings yet to be fully explored in the literature. Further issues with the CICE coupling are discussed in the appendix.

This CICE anecdote drives at my main criticism of this paper as it currently stands: It seems to be a vacant conclusion to discuss model efficiency without discussing model accuracy. The most efficient model one can design is a constant number, but seldom is this model the most accurate. The only way this limitation in the current manuscript can be remedied is to explicitly state the configurations used for each particular model in the tests presented, including graphically representing the domains used. However, due to the number of different models and model configurations used, this may balloon the paper to unmanageable proportions. However, as the paper currently stands, there is too little information available for it to be useful for other groups trying to address coupled model efficiency in their particular configurations.

Conclusion

In some respects, the scope of this paper is too large and should be refined. The concluding arguments would be far more compelling, and, I believe, interesting to the modeling community, if it explored individual coupled configurations, and efficiency related to a group of relatively standard model settings in each component model. However, this is probably beyond the scope intended by the authors, and therefore one way to make sure the good work already done is published would be to: 1) Provide greater details of each of the models used to produce the results, including model domain maps,

of the model configuration tables, the latter in an appendix; and 2) Provide at least some indication of the accuracy of the solutions. Otherwise, one is left to wonder as to how exactly the results were produced. Currently the paper fails the reproducibility test , because insufficient information is provided to repeat the experiments. This, alone, is grounds for significant revision, which I hope the authors will undertake.

—-

Appendix – CICE configuration and coupling

This appendix addresses technicalities of the CICE setup that were puzzling to the reviewer. First, the authors may be interested to know that there were important bug fixes in the code between version 5.0 and 5.1 of CICE (update is in Hunke et al., 2015), however these would be unlikely to influence computational performance. Setting this aside, there are further improvements in the computational performance of the model using EAP that are being updated by the University of Reading at the current time. It is impossible to know whether or not this affects the results in this paper, because the CICE configuration used in this paper is never made clear. Also, and perhaps I missed it in the text, whether or not the namelist option "distribution_type" is changed in CICE is not discussed. This affects computational performance.

Most importantly, however, is the information within Table 5 on how CICE is coupled to CCLM. My understanding is that the âŁŮ symbol indicates fluxes being passed from CCLM to CICE. If this is the case, there is only one feedback from CICE to CCLM in Table 5 (SST), which draws into question the physical consistency of the coupling. If this were to be a fully coupled model, then there must be more feedbacks that just surface temperature to the atmosphere. For sea ice, the most important feedback is either albedo or reflected shortwave radiation, passing back from the sea ice model to the atmosphere, but neither is listed, which leads one to assume that albedo is being calculated in the atmospheric model independently. Given the sophistication of the Delta-Eddington albedo parameterization in CICE, this seems odd. This inconsistency

should be addressed before publication.

It is also odd that the atmosphere is calculating sensible and latent heat fluxes, given that the CICE configuration has five sea ice thickness categories each calculating an independent surface temperature upon which turbulent fluxes are based. Hence the turbulent heat fluxes must be inconsistent with the surface stress term, which is being calculated internally in CICE in the configuration given. When this calculation is done within CICE, assuming Monin-Obukhov stability calculations are being performed, the drag coefficient accounts for the individual surface temperature of each of the five sea ice thickness categories. If this calculation is not being performed in CICE, then the only alternative would be for the sea ice model to use only neutral drag, which would also be inconsistent with the sensible and latent heat flux components of turbulent transfer being passed from the atmosphere. The only way to remedy this is either to specify surface stress from the atmospheric model, or to fully use the turbulent transfer calculations in CICE, and pass the sensible and latent heat fluxes back to the atmosphere from the sea ice model. This is the reverse of what is currently being done, or at least described in this manuscript. This inconsistency should also be addressed before publication.

References

Craig, A. P., M. Vertenstein, and R. Jacob (2012), A new flexible coupler for earth system modeling developed for CCSM4 and CESM1, Int. J. High Perform. Comput. Appl., 26(1), 31–42, doi:10.1177/1094342011428141.

---

## Referee Comment (RC2) · S. Valcke (Referee) · 8 Jun 2016

General comments

This paper present a detailed analysis of the performance of coupling configurations involving the COSMO-CLM regional model. An extensive literature exists on the performance analysis of individual models or codes but there is much less published on the performance of the coupled system and on the coupling aspects per se. This paper addresses this gap and as the work onto which it is based is sound, it deserves publication. However, I consider it needs major revisions before being published.

[Figure]
Specific comments

My first main concern is about the way the results on the optimum configurations (p.25, section 4.4, Fig. 5 and Table 8) are presented ; currently, they are difficult to appreciate because there is, on one hand, a lot of information (sometimes superfluous), and on the other hand, some missing details.

1. First of all, I do not understand why the cost of the CCLM part of CCLM-CLM and CCLM+VEG3D are about doubled compared to the costs of the CCLM stand alone or compared to the CCLM-NEMO-MED12 coupling. On p.25, l.854, it is stated "The corresponding costs are about double the costs of the stand-alone reference: 512.0 and 473.6 CHPSY, respectively". Can you give an explanation? Is it linked to the fact that CCLM runs in SMT non-alternating mode in stand-alone and in the CCLM-NEMO-MED12 coupling where as it runs in SMT alternating mode in the CCLM-CLM and CCLM+VEG3D couplings? If so, it should be stated in the text.

2. The fact that the COSMO version used for CCLM+CLM is different from the COSMO version used for CCLM+VEG3D and that the results presented for CCLM+VEG3D are in fact not the optimum ones (128 cores were chosen to be able to compare with CCLM-CLM) is disturbing and the paragraph p.25 l-843-855 is difficult to understand (same thing for p.27, l922-923). I am not sure on how to correct this but this should be simplified maybe simply by removing results for CCLM-CSM?

3. It is not clear on figure 5 if the time to solution includes or not the OASIS interpolations. Can you clarify this? It is written in Table 8 captions that it does not but it should be stated either in Fig 5 captions and in the text, stressing that the interpolation time is relatively small anyway (as quantified in Table 8). Can you also specify that the OASIS interpolation times are provided directly by the lucia tool in table 8 captions (even if this is mentioned in the text on p.21)

My second concern is about the definition of the criteria to identify the optimum configuration, which are not clear:

[Figure]

4. In section 4.2, please specify what you mean by "each component's gain in speed, compared to its speed on one node, outweighs the increase in costs." The units of speed are not the same than the units of cost so they cannot be compared directly. Are you considering the relative gain in speed (in %) is compared to the relative increase of cost (in %)? To help the understanding, one practical example with numbers should be given (maybe at the beginning of current section 4.4?), for example the steps that lead to the identification of the optimum configuration for the CCLM

5. I think section 4.2 would be better to place juste just before 4.4 (i.e changing switching current 4.2 and 4.3 sections)

6. p.22, l.749: This constraint is effective only for sequential coupling so it should be moved to the paragraph currently starting line 754.

7. p.22 l.763 & l.779: It is not clear who or what decides if the costs are a limiting factor or not. Did you consider the costs was a limiting factor in your identification of the optimum? If so, what was the limit? This should be clarified.

My third major concern is about section 4.5.2

8. I find this section not really relevant in the context of this paper. Of course, one can always get better fictive results by neglecting costly or badly written parts of the code!

Then I have the following major remarks:

9. In general, I think the text is quite heavy with many repetitions. I suggest to make it lighter and more "right-to-the-point". In particular, section 2 describing the components could be reduced and the appendix A, that is not essential to the understanding of the paper could be given as supplementary material.

10. With which version of COSMO were the CCLM stand-alone tests done? Is it cosmo_4.8_clm19 like for all coupling but CCLM-CLM? This should be clarified in the text.

11. p.3, l.69: Please give some details on why the MESSy approach was not considered.

12. p.5, l.597: You mention a "coupling weight" increasing to 1 with time but this coupling weight is not described. Can you explain with more details how it works?

13. p.21, l.724: can you justify the formula used to approximate the time for 40 levels based on the time for 45 levels; why not simply use : T40 = T45 x 40/45

14. p.24, 806-807: I do not understand how one can conclude that "COSMO-CLM in ST and SMT mode exhibits a very similar PE for the same number of processes ..." The curves are distinct. Do you mean that we should compare the SMT results for a specific number of cores with the ST results with twice as many cores (to get to the same number of processes)

15. p.24, l.808: you write "an increased loss of PE between 160 and 80 grid points per process." but the reader cannot directly infer the number of grid points per process given the number of cores (which is the information provided on the figure), so the corresponding number of cores should be mentioned to help the reader.

16. p.24, l.813-814: I do not fully understand this sentence. First I am not sure what the "component interface" is. Is it the coupling interface? If so, I do not understand how to reconcile this with the fact that the coupling interface time probably includes the time for interpolations (which are done either on the source side before the sending or on the target side after the receiving) and that the "time to solution" does not.

17. p.24, l.815-819: I do not understand the meaning of the sentence "Hereby, the number of cores and the threading mode (ST or SMT) are kept constant." I propose to remove this sentence and rewrite the following ones as: " COSMO-CLM components of concurrent couplings should be compared to stand-alone COSMO-CLM in SMT mode because in both cases two threads per core are used to run COSMO-CLM. Conversely, COSMO-CLM components of sequential couplings should be compared to stand-alone

COSMO-CLM in ST mode because in both cases only one thread per core is used to run COSMO-CLM." if I am right in my interpretation.

18. p.24, l.826-827: It is written "However, as mentioned in section 2.6 CLM is coupled to cosmo_5.0_clm1 model version which is a more recent version than cosmo_4.8_clm19 used for all other couplings " but I don't see this mentioned in section 2.6.

19. p.26, l.898-900: I propose to rephrase these two sentences for "It is not surprising that the couplings with soil-vegetation models shows only moderate extra costs as they replace the use of TERRA, the internal soil-vegetation model activated in stand-alone versions of COSMO-CLM. "

20. p.28, l.949-955: This paragraph is not clear. Going from non-alternating to alternating reduces the time to solution by 35.1 %. Improving the performances of the derivative calculation reduces the time to solution by 9.2%. Going from 16 cores in SMT mode to 32 cores in ST mode results in a reduction of time to solution by 25.5 %. But then why is the "discrepancy" calculated by comparing this 25.5% to the 9.2% linked to the improvement of the derivative calculations? It should be calculated by comparing the 25.5% to the non-alternating to alternating gain of 35.1%, shouldn't it?

21. p.30, l.1040-1043: The 10% variation in the time to solution results should be introduced in the text and not only in the conclusion.

Minor remarks and technical corrections

22. I think it would be less confusing to use CCLM everywhere and not sometimes CCLM and sometimes COSMO-CLM

23. p.1, l.8: The OASIS3-MCT interface is not really described in the paper. I suggest changing "present" for "use"

24. p.3, l.58 & p.6, l.166: Valcke 2013 refers to a paper describing the "old" OASIS3 version and not the more recent OASIS3-MCT version. The reference Valcke et al.,

2013 should be used instead.

25. p.3, l.67: I propose changing "is based" for "would be based"

26. p. 4, l.94: Please add "depends" after "but" in "but on the coupling method

27. p.6, l.168: Please add "which" after "data" in "amount of data is a requirement"

28. p.6, l.186-187: The sentence "The coupling of COSMO-CLM with the global ocean model NEMO is realized by means of two different regional versions of the NEMO model ..." sounds weird to me because of the opposition between "global" and "regional". I suggest simply "COSMO-CLM is coupled to two different regional versions of the NEMO model ..."

29. p.10, l.337: The fact that each component needs to be a separate executable is not a constraint anymore with the last OASIS3-MCT_3.0 version; maybe this could be mentioned.

30. p.11, l.339 & l.366: Please change "whose" for "which"

31. p.11, l.343: I suggest changing "is directly executed via the Message Passing Interface" for "is directly executed via the Model Coupling Toolkit (MCT, Jacob et al 2005) based on the Message Passing Interface (MPI)" and add the reference "Jacob, R., J. Larson, and E. Ong: MxN Communication and Parallel Interpolation in CCSM3 Using the Model Coupling Toolkit. Int. J. High Perf. Comp. App., 19(3), 293-307 2005 "

32. P.11, l.357-358: I suggest changing "This component partitioning does not have to be the same" for "The component partitioning and grid do not have to be the same"

33. p.11, l.361: I suggest adding "and accumulation" after "time averages"

34. p.11, l.373: I suggest changing "OASIS3-MCT includes the MPI library" for "OASIS3-MCT includes the MCT library based on MPI " (but this is redundant with p.11, l.343 -see also my remark #9 above

35. p.13, l.428: Please add a ) after 4.1

36. p.13, l. 442: Please change "interpolation" for "coupling" as it is not only the interpolation that is improved but the interpolation and the communication.

37. p.20, l.687-691: I am not convinced these are effectively the two main goal of performance analysis. These sentences are unnecessary and contribute to the heaviness of the text (see also my first "Important remark" above.

38. p.21, l.722: Please change "compansated" for "compensated"

39. p.22, l.737-738, I suggest rephrasing the sentence "In a perfectly scaling parallel application the costs would remain constant if the resources are doubled, the parallel efficiency would be 100 %, the speed would be doubled and the speed-up would be 200 %. " for "If the resources of a perfectly scaling parallel application are doubled, the speed would be doubled and therefore the cost would remain constant, the parallel efficiency would be 100 %, and the speed-up would be 200 %."

40. p.23, l.791: Please change "CPUh" for "core hours" to be coherent with the rest of the text.

41. Table 8 should be placed after Table 7 and not after all the Figures.

42. p.26, l906: Please change "atmosphere" for "coupled model"

43. p.30, l.1031: Please change "scaling;" by "scaling,"

---

## Author Comment (AC1) · 20 Jan 2017

The number 4.8 is added in the title of the revised version.

---

## Author Comment (AC2) · 20 Jan 2017

Dear Colleague, we would like to thank you for all your comments, remarks and questions and we considered all of them carefully. However, ff we understand you correctly, we fear, you will not be happy with all of our answers and with all aspects of the article revision.

We use model configurations which are regarded to be optimum for Europe and investigate an optimum configuration of resources in coupled mode by keeping the configurations of model physics and dynamics unchanged. The physics of the coupling is

described only. An analysis of its accuracy and its scientific discussion is not the topic of this paper. We hope having made it more clear in the article and in our answer to your review.

It is also beyond the scope of the paper to investigate the computational efficiency in dependency on the models accuracy. The prerequisite of such an investigation would be the quantificiation of models accuracy in a more theoretical way than by comparison of the results with observations in order to make such a study feasible.

We are looking forward to your comments on our revision and remain with

kind regards, Andreas Will

Please also note the supplement to this comment:
http://www.geosci-model-dev-discuss.net/gmd-2016-47/gmd-2016-47-AC2-supplement.pdf

**Supplement:**

**Response to GMD-2016-47-RC1 (anonymous)**

**Introduction**

Coding and technical aspects of coupling Earth System Models are often relegated to institutional reports seldom referenced or widely read, and outcomes of work in coupling and load balancing are often blindly used by physical and biogeochemical modeling groups. Therefore I commend the authors for documenting their expansive coupling work, and for submitting it to be reviewed for a journal with a readership that bridges the coupler development and physical modeling communities. I do, however, have reservations about the final results, methods, and one comment about the scope of the cited literature.

**Answer:** we appreciate that the effort we made to document our technical work could be hosted by the journal. In this revision, we try to answer the referee questions, adding details, without overloading the article too much.

**Significance of this paper in the context of other work**

The manuscript's introduction could extend the perceived reach of the work if it were to illustrate its international significance. There are several latest-generation regional coupled earth system models in development in the U.S. and Canada, some of which use MCT and takes advantage of the work of Craig et al. (2012) that could have been cited and have appeared in the reviewed literature in recent years. The reason I mention these publications is to say that the introductory argument perhaps could be further enhanced, since work on load balancing high-resolution regional coupled earth system models is taking place in many parts of the Earth System Modeling community. This helps to widen the appeal of the current manuscript, and its significance.

**Answer:** the referee rightly emphasises that load balancing is not a new issue in our community. Studies based on CESM model, for example, are familiar to the authors (Craig 2012 is cited in the article, line 385). We have also mentioned Dennis et al. 2012 (line 88) and Alexeev at al. 2014 (line 90) in our introduction. We added a reference to Balaprakash et al 2014 (line 809) in chapter "4.3 Strategy for finding an optimum configuration" and we have mentioned that, in our case, "due to the heterogeneity of our coupled systems, a single algorithm cannot be proposed (as in Balaprakash et al, 2014)". Unless the CESM package, the OASIS library allows an unlimited kind of component combination in coupled systems. For the moment, it is rather complicated to propose an automatic load balancing tool that could deliver an optimal solution for all combinations. We hope that the present article will help the OASIS community to develop an ability to better balance their systems and, in a second step, propose solutions that may be gathered in a single tool.

**Efficiency versus accuracy**

This paper discusses a considerable number (five) of different coupled model configurations using CCLM, however only scant information is provided on each one of these configurations. It would be particularly useful to view maps of model domains to demonstrate the individual configurations for each of the coupled model systems in Table 2. Answer: Model domains are shown now in Figure 1.

This would help make it clear exactly how much ocean, land and sea ice exist in the respective model domains. Such details can have a large impact on scalability and parallel efficiency, especially in the cryosphere (sea ice and snow). Therefore I suggest providing greater detail on the physical configuration of each of the models chosen, because this, too, has an enormous impact on the model solution.

Answer: We fully agree with the reviewer that details on the physical configuration have an impact on individual performances of components, and consequently, on performances of the whole coupled system. However, the article is not investigating the physical

2

performance of the coupled systems. It is rather focussing on presentation of the coupling method, the computational performance in COSMO-CLM reference configuration and finding of an "optimum configuration". It is thus out of scope to discuss in detail the amount of ice and snow in the model domain and the impact on computational performance. It is also out of scope to discuss physical and dynamical parameters that could influence the computing performance. This remains for future work. Further below we propose an improved definition of our component characteristics by parameters relevant for computing performances. We also answer the referee's questions about CICE.

To illustrate this point, I focus here on the implementation of CICE Version 5 for CCLM+TRIMNP+CICE. The computational efficiency of the solution in CICE is heavily dependent upon the total number of sea ice thickness categories used, the number of tracers needed, for example, by melt pond and ice-age tracking and biogeochemistry, and most importantly, the sea ice mechanics solution. If CICE 5 has been configured to use anisotropic (Elastic Anisotropic Plastic; EAP) sea ice mechanics, then it will definitely be expensive, and, could take as much as 30% of the total model execution time in pan-arctic fully coupled regional models, if a highly converged plastic sea-ice solution is required (2second sub-cycling). However, if using the Elastic Viscous Plastic (EVP) sea ice rheology with 10-second sub-cycling, the time to solution of the sea ice model greatly improves, with only slight degradation of the plastic solution. In this configuration, the sea ice model could take only 10% of the total core time of running the model. It is still unknown as to which of the two variants is physically more accurate. This is precisely the same CICE Version 5.1 code, in the same coupled framework, using MCT, but with two different namelist settings yet to be fully explored in the literature. Further issues with the CICE coupling are discussed in the appendix.

**Answer:** EVP was used (kdyn=1). However, CICE domain covers only the Baltic Sea and Kattegat, not the pan-arctic. The sea ice which appears in a relatively small domain like the Baltic Sea and disappears totally in summer has less complicated features compared to the Arctic. However, we cannot say how much different the calculations would be if EAP was chosen, as no sensitivity tests about these parameters have been conducted. The scope of the paper was to present a strategy of analysis of the computational performance of the coupled system in comparison to stand-alone performance. A deeper analysis is out of scope of the paper and remains for future work. We highlight the relevance and the opportunities of such an analysis in the result section for the CCLM+MPI-ESM coupling (line 1111 ff).

This CICE anecdote drives at my main criticism of this paper as it currently stands: It seems to be a vacant conclusion to discuss model efficiency without discussing model accuracy. The most efficient model one can design is a constant number, but seldom is this model the most accurate. The only way this limitation in the current manuscript can be remedied is to explicitly state the configurations used for each particular model in the tests presented, including graphically representing the domains used. However, due to the number of different models and model configurations used, this may balloon the paper to unmanageable proportions. However, as the paper currently stands, there is too little information available for it to be useful for other groups trying to address coupled model efficiency in their particular configurations.

**Answer:** The aim of the paper is to analyse the performance of the coupled systems using a configuration common for climate applications. Therefore, the analysis of computational performance was conducted using well tested and recommended climate modelling configurations for each component model without any idealisation, e.g. the I/O is the same as in standard climate applications. This is described in section 4.1, line 746 ff.. We agree with the reviewer that a detailed description of all configuration would balloon the paper and

hope having found an appropriate compromise concentrating on configuration details specific for the couplings described in chapter 2.

However, the computing performances of the coupled system necessarily depends on the performances of each component. We agree with the referee that the choice of an additional component cannot only depend of its computing cost. Obviously, the model accuracy (or model skill) is the most important criterion. The article does not say anything about component accuracy in stand alone mode or, what we consider to be even more important, component accuracy in coupled mode. This article addresses the usability of configurations, which is a prerequisite of scientific analysis described in other papers, as for example Pham et al. 2016 (CCLM+NEMO-NORDIC) or Davin et al. 2016 (CCLM+CLM).

Nevertheless, we agree that more information is usefull to facilitate the comparison of component costs and to estimate the cost of possible other configurations (e. g. with other resolutions). An interesting suggestion is the computing performance metrics described in Balaji et al. 2017, particularly the 2 parameters describing the models: resolution and complexity. "Resolution" -G- is measured as the number of grid points (or more generally, spatial degrees of freedom) NX,NY,NZ per component. "Complexity" -V- is measured as the number of 3D prognostic variables per component (to be able to compare 3D models, like atmosphere, with 2D models, like land models, it is assumed that V of 2D models are equal to 1). These 2 parameters are added in Table 3.

G and V are key parameters to explain why some components are more costly than others (MPI-ESM, with highest G and V, is also the one which induces the highest coupling cost). This information is emphasised in § 4.5 "Extra time and costs", line 1005 ff. It can also be used for users who would like to estimate the extra cost induced by changes in a coupled component, like a resolution increase (horizontal or vertical) or a complexity increase (additional calculations like biogeochemistry in the ocean or chemistry in the atmosphere ...)

**Conclusion**

In some respects, the scope of this paper is too large and should be refined. The concluding arguments would be far more compelling, and, I believe, interesting to the modeling community, if it explored individual coupled configurations, and efficiency related to a group of relatively standard model settings in each component model. However, this is probably beyond the scope intended by the authors, and therefore one way to make sure the good work already done is published would be to: 1) Provide greater details of each of the models used to produce the results, including model domain maps, of the model configuration tables, the latter in an appendix; and 2) Provide at least some indication of the accuracy of the solutions. Otherwise, one is left to wonder as to how exactly the results were produced.

**Answer:** As already stated, (1) we use recommended and overall tested model configurations for climate application over Europe. (2) a map is added in Figure 1 showing the model domains and (3) metrics are added in Table 3 to better estimate the model accuracy. Furthermore, the results of computational performance are revised and presented in a more consistent way. Figures 5 and 6 together with table 8 provide consistent results. In table 8 the section 3.3 shows a systematic analysis of extra costs of coupling for all couplings investigated at optimum configuration. The components are described in lines 920 ff.

Currently the paper fails the reproducibility test, because insufficient information is provided to repeat the experiments. This, alone, is grounds for significant revision, which I hope the authors will undertake.

**Answer:** We thank the reviewer for this comment. The reproducibility of results is an important aspect of community work in the CLM Community and we hope to be able to show it in the following. We added details on how to get the model versions and configurations used for the performance analysis presented in the Appendix under "source code availability", line 1135 ff. At the moment, the model versions used are not official CLM-Community model versions but available from the model developers. An implementation into an official CLM-Community released version is ongoing. Hereby we follow the procedure of source code development introduced in the COSMO and CLM Community. Each experiment can be repeated with the set up information from the article and using the model input files. To get the individual coupled systems, model input files and configuration details the authors have to be contacted as described in the Appendix.

All results presented and the original model output files used are available from the lead author, following the rules of good scientific practice.

However, the machine blizzard is not available anymore. Thus the results are, strictly speaking, not reproducible. This, however, is not the responsibility of the authors and true for each numerical model result after some years. The authors believe that the results highlighted are robust and can be obtained on a similar machine as well.

**Appendix – CICE configuration and coupling**

This appendix addresses technicalities of the CICE setup that were puzzling to the reviewer. First, the authors may be interested to know that there were important bug fixes in the code between version 5.0 and 5.1 of CICE (update is in Hunke et al., 2015), however these would be unlikely to influence computational performance. Setting this aside, there are further improvements in the computational performance of the model using EAP that are being updated by the University of Reading at the current time. It is impossible to know whether or not this affects the results in this paper, because the CICE configuration used in this paper is never made clear. Also, and perhaps I missed it in the text, whether or not the namelist option "distribution\_type" is changed in CICE is not discussed. This affects computational performance.

**Answer:** Parameters used in CICE and TRIMNP are the same as in real climate simulations for Europe. They are listed and discussed in the following but not included in as much detail in the paper.

**CICE:**

- + kitd = 1; ktherm = 2; conduct = 'MU71'
- + kdyn = 1 (means EVP is used); ndte = 60; revised\_evp = .false.; advection = 'upwind'
- + shortwave = 'dEdd'; albedo\_type = 'default'
- + tr\_brine = .false.; skl\_bgc = .false.; bgc\_flux\_type = 'Jin2006'
- + formdrag = .false.

+ tr\_iage = .true.; tr\_FY = .true.; tr\_lvl = .true.; tr\_pond\_cesm = .false.; tr\_pond\_topo = .false.; tr\_pond\_lvl = .true.; tr\_aero = .false.

+ distribution\_type = "cartesian"; processor\_shape = "square-pop"; distribution\_wght = "latitude"; ew\_boundary\_type = "open"; ns\_boundary\_type = "open"

The original formula of category boundary (kcatbound = 0) with the thickness boundaries for five thickness categories and the linear remapping of the ice thickness distribution (kitd = 1) are configured in this study. The thermodynamics option new "mushy" formulation (ktherm=2) is applied in which salinity evolves (Turner et al., 2013). For each thickness category, CICE computes changes in the ice and snow thickness and vertical temperature profile resulting from radiative, turbulent, and conductive heat fluxes. The ice has a temperature-dependent specific heat to simulate the effect of brine pocket melting and freezing. The standard thermal conductivity option used is 'MU71' following Untersteiner (1964) and

The distribution type option is the standard Cartesian distribution of blocks which allows redistribution via a 'rake' algorithm for improved load balancing across processors, and redistribution based on space-filling curves. The processor shape is square-pop. The 'latitude' option weights the blocks based on latitude and the number of ocean grid cells they contain. The Neumann boundary conditions are set up for both east-west and north-south boundary type.

**TRIMNP:**

hdif\_u=50., hdif\_v=50., hdif\_w=0., hdif\_s=25., hdif\_t=25., hdif\_q=0.,

The dynamics of the free surface are discretised semi-implicitely, and the resulting linear equation system is solved with a pre-conditioned conjugate gradient method. The vertical mixing and friction including non-linear bottom friction and surface wind stress are also solved with a semi-implicit method. The vertical mixing and friction coefficients are parameterised using prognostic equations for turbulent kinetic energy and dissipation (Umlauf and Burchard 2005). For horizontal diffusion, harmonic terms are used with scale dependent constants. The lateral diffusion and the viscosity constants are 25 m2/s and 50 m2/s, respectively. Advection for all time-dependent variables is done with a Semi-Lagrangian method, where at the end of each time step the values of the variables at the corresponding grid points (the arrival points) are determined by following a trajectory backwards in time for one time step interval to the departure points. The values of the variables at the variables at the departure points are determined by trilinear interpolation. For details see Cheng et al. (1993).

Most importantly, however, is the information within Table 5 on how CICE is coupled to CCLM. My understanding is that the U symbol indicates fluxes being passed from CCLM to CICE. If this is the case, there is only one feedback from CICE to CCLM in Table 5 (SST), which draws into question the physical consistency of the coupling. If this were to be a fully coupled model, then there must be more feedbacks that just surface temperature to the atmosphere. For sea ice, the most important feedback is either albedo or reflected shortwave radiation, passing back from the sea ice model to the atmosphere, but neither is listed, which leads one to assume that albedo is being calculated in the atmospheric model independently. Given the sophistication of the Delta-Eddington albedo parameterization in CICE, this seems odd. This inconsistency should be addressed before publication.

It is also odd that the atmosphere is calculating sensible and latent heat fluxes, given that the CICE configuration has five sea ice thickness categories each calculating an independent surface temperature upon which turbulent fluxes are based. Hence the turbulent heat fluxes must be inconsistent with the surface stress term, which is being calculated internally in CICE in the configuration given. When this calculation is done within CICE, assuming Monin-Obukhov stability calculations are being performed, the drag coefficient accounts for the individual surface temperature of each of the five sea ice thickness categories. If this calculation is not being performed in CICE, then the only alternative would be for the sea ice model to use only neutral drag, which would also be inconsistent with the sensible and latent heat flux components of turbulent transfer being passed from the atmosphere. The only way to remedy this is either to specify surface stress from the atmospheric model, or to fully use the turbulent transfer calculations in CICE, and pass the sensible and latent heat fluxes back to the atmosphere from the sea ice model. This is the reverse of what is currently being done, or at least described in this manuscript. This inconsistency should also be addressed before publication.

**Answer:** We agree that the inconsistency exists and needs to be improved in the future. We explain this inconsistency in the paper now (chapter 3.4, line 630 ff). In the experiment CCLM+TRIMNP+CICE, only SSTs are passed to the atmosphere as in the version of CCLM used at the time when the experiment was conducted for this study the partial sea ice cover, snow on sea ice and water on sea ice are not considered. In a water grid box of CCLM, the albedo parameterisation switches from ocean to sea ice if surface temperature is below a freezing temperature threshold of -1.7°C. We would have passed sea ice fraction to CCLM as it was done for NEMO-Nordic. However, we think that careful checks e.g. for reflected shortwave radiation should be made for the coupled system model CCLM+-TRIMNP+CICE if sea ice fraction and albedo from CICE are sent to CCLM. These checks remain for future work.

In the current study, no sea ice information from CICE was passed to CCLM. But they were sent to TRIMNP. In TRIMNP the surface temperature is calculated as a combination of SSTs from TRIMNP and the sea ice skin temperatures from CICE, weighted by the sea ice concentration before the combined surface temperature is passed to CCLM. In Table 5, "surface temperature over sea/ocean" is used instead of SST to avoid a potential misunderstanding in case of sea ice existence.

We also think that even if sea ice fraction from CICE is sent to CCLM, the latent and sensible heat fluxes in CCLM are still different to those in CICE due to different turbulent schemes of the two models CCLM and CICE. The inconsistency can be removed only if all models use the same energy fluxes, calculated in one model at the highest resolution, for example in CICE model, as the reviewer suggested. This strategy could be applied in future studies considering the result of this performance study, that exchanging much more fields has a small impact on cost.

**References:**

• Alexeev, Y., Mickelson, S., Leyffer, S., Jacob, R., and Craig, A., 2014: The Heuristic Static Load-Balancing Algorithm Applied to the Community Earth System Model, in: 28th IEEE International Parallel and Distributed Processing Symposium, no. 28 in Parallel & Distributed Processing Symposium Workshops, pp. 1581–1590, IEEE, doi:10.1109/IPDPSW.2014.177

• Balaji, V., Maisonnave, E., Zadeh, N., Lawrence, B. N., Biercamp, J., Fladrich, U., Aloisio, G., Benson, R., Caubel, A., Durachta, J., Foujols, M.-A., Lister, G., Mocavero, S., Underwood, S., and Wright, G., 2017: CPMIP: Measurements of Real Computational Performance of Earth System Models in CMIP6, *Geosci. Model Dev.*, **46**, 19-34, doi:10.5194/gmd-10-19-2017

• Balaprakash, P., Alexeev, Y., Mickelson, S. A., Leyer, S., Jacob, R. L. and Craig, A. P., 2014: Machine learning based load-balancing for the CESM climate modeling package, in Proceedings for 11th International Meeting on High-Performance Computing for Computational Science (VECPAR 2014)

• Davin, E. L., Maisonnave E. and Seneviratne, S. I.,2016: Is land surface processes representation a possible weak link in current Regional Climate Models ? , *Environ. Res. Lett.*, **11** 074027

• Dennis, J. M., Vertenstein, M., Worley, P. H., Mirin, A. A., Craig, A. P., Jacob, R., and

Mickelson, S., 2012: Computational performance of ultra-high-resolution capability in the Community Earth System Model, Int. J. High Perf. Comp. Appl., 26, 5–16, doi:10.1177/1094342012436965

• Cheng, R., Casulli, V., Gartner, J., 1993. Tidal, Residual, Intertidal Mudflat (TRIM) Model and its Applications to San Francisco Bay, California. Estuarine, Coastal and Shelf Science, 36, 235–280

• Jin, M., Deal, C. J., Wang, J., Shin, K. H., Tanaka, N., Whitledge, T. E., Lee, S. H., and Gradinger, R. R., 2006. Controls of the landfast ice-ocean ecosystem offshore Barrow, Alaska. Ann. Glaciol., 44:63–72.

• Maykut G. A. and Untersteiner N., 1971. Some results from a time dependent thermodynamic model of sea ice. J. Geophys. Res., 76:1550–1575.

• Pham, Trang Van, J. Brauch, B. Früh, B. Ahrens, 2016: Simulation of snowbands in the Baltic Sea area with the coupled atmosphere-ocean-ice model COSMO-CLM/NEMO.*Met. Z.*, DOI: 10.1127/metz/2016/0775

• Souverijns, N., Gossart, A., Demuzere, M., Lhermitte, S., Gorodetskaya, I., Van Lipzig, N., 2016: Evaluation of a default COSMO-CLM simulation over Antarctica with a focus on accumulation and the surface mass balance, Cosmo Assembly, Lüneberg, 20-23 September 2016

• Turner, A. K., Hunke, E. C., and Bitz, C. M., 2013. Two modes of sea-ice gravity drainage: a parameterization for large-scale modeling. J. Geophys. Res., 118:2279–2294, doi:10.1002/jgrc.20171.

• Umlauf, L., Burchard, H., 2005. Second–order turbulence models for geophysical boundary layers. A review of recent work. Cont. Shelf Res., 25, 795–827.

• Untersteiner, N., 1964. Calculations of temperature regime and heat budget of sea ice in the Central Arctic. J. Geophys. Res., 69:4755–4766.

---

## Author Comment (AC3) · 20 Jan 2017

Dear Ms. Kerkweg,

thank you for your comment.

The availability of all components is specified in Appendix A of the revised version submitted today as "source code availability". The sentence related to COSMO-ART was removed.

Kind regards, Andreas Will

---

## Author Comment (AC4) · 20 Jan 2017

Dear Dr. Valcke,

we would like to thank you for your detailed review. We considered all of your suggestions, remarks, critical comments and questions. We hope having found a satisfactory solution in the revised manuscript.

We kept the structure of your review and answer the points consequetively. Within the answer you may find a hint to the section, figure and/or line number in the revised manuscript.

[Figure]

However, we changed some other parts of the manuscript as well in order to improve the readability of the text, in particular chapter 4 "Computational Efficiency". Thus, we would like to suggest to read this chapter first to get an impression of the new result presentation before investigating our detailed answers to your review points. The abstract and the conclusions are changed accordingly.

We are looking forward to your comments on the revised version and remain with

kind regards,

Andreas Will

Please also note the supplement to this comment:
http://www.geosci-model-dev-discuss.net/gmd-2016-47/gmd-2016-47-AC4-supplement.pdf

───────────────────────

**Supplement:**

**Response to GMD-2016-47-RC2 (Sophie Valcke)**

**Dear Ms. Valcke,**

we thank you for the constructive and detailed comments and questions and hope to give an easy to follow and satisfactory answer. We tried to consider all of your points and some more with the article revision. We also kept some redundancy of the basic aims in order to facilitate following the idea of the article.

Please, keep in mind that all references to the paper given in the following are references to the revised version of the article. Your comments are given in black, our answers in blue.

**Best regards, Andreas Will**

**General comments**

This paper present a detailed analysis of the performance of coupling configurations Involving the COSMO-CLM regional model. An extensive literature exists on the performance analysis of individual models or codes but there is much less published on the performance of the coupled system and on the coupling aspects per se. This paper addresses this gap and as the work onto which it is based is sound, it deserves publication. However, I consider it needs major revisions before being published.

**Answer:** We are very pleased to know that our paper is interesting from your point of view and we did the best to answer your questions.

**Specific comments**

My first main concern is about the way the results on the optimum configurations (p.25, section 4.4, Fig. 5 and Table 8) are presented ; currently, they are difficult to appreciate because there is, on one hand, a lot of information (sometimes superfluous), and on the other hand, some missing details.

**Answer:** Thank you for that comment. We revised chapter 4, improved the figure description, separated figures 3-4 "time to solution" and "cost" from figure 5 "parallel efficiency", which belongs to finding of optimum configuration, we introduced a separation of extra cost in 5 components and revised table 8 accordingly considering the reviewer comments. In particular, the last section 3.3 of table 8 is revised and all numbers are presented in a consistent and unique way as % of cost of optimum configuration of CCLM stand-alone. We removed the figures and the discussion of possible improvements of CCLM+MPI-ESM.

First of all, I do not understand why the cost of the CCLM part of CCLM-CLM and CCLM+VEG3D are about doubled compared to the costs of the CCLM stand alone or compared to the CCLM-NEMO-MED12 coupling. On p.25, I.854, it is stated "The corresponding costs are about double the costs of the stand-alone reference: 512.0 and 473.6 CHPSY, respectively". Can you give an explanation? Is it linked to the fact that CCLM runs in SMT non-alternating mode in stand-alone and in the CCLM-NEMO-MED12 coupling where as it runs in SMT alternating mode in the CCLM-CLM and CCLM+VEG3D couplings? If so, it should be stated in the text.

**Answer:** We agree with the reviewer that it is difficult to follow the discussion and improved it (hopefully). You find a new paragraph (section 4.5. line 927 ff) clarifying which the dominating components of extra cost of coupling are. It turned out, it is mainly due to using ST instead of SMT mode and of the double number of cores.

"On p.25, I.854, it is stated "The corresponding costs are about double the costs of the stand-alone reference: 512.0 and 473.6 CHPSY, respectively". Can you give an explanation? Is it linked to the fact that CCLM runs in SMT non-alternating mode in stand-alone and in the CCLM-NEMO-MED12 coupling where as it runs in SMT alternating mode in the CCLM-CLM and CCLM+VEG3D couplings?"

Answer: You are right to a wide extend. See previous answer.

The fact that the COSMO version used for CCLM+CLM is different from the COSMO version used for CCLM+VEG3D and that the results presented for CCLM+VEG3D are in fact not the optimum ones (128 cores were chosen to be able to compare with CCLM-CLM) is disturbing and the paragraph p.25 I-843-855 is difficult to understand (same thing for p.27, I922-923). I am not sure on how to correct this but this should be simplified maybe simply by removing results for CCLM-CSM?

**Answer:** Thank you for the comment. We conducted additional measurements comparing cosmo\_5.0\_clm1 used in CCLM+CLM and cosmo\_4.8\_clm17 used in CCLM+VEG3D (on another machine since blizzard is not availale anymore). This exhibited 45% higher cost of 5.0. We revised the result presentation and in particular this paragraph .

It is not clear on figure 5 if the time to solution includes or not the OASIS interpolations. Can you clarify this? It is written in Table 8 caption that it does not but it should be stated either in Fig 5 captions and in the text, stressing that the interpolation time is relatively small anyway (as quantified in Table 8). Can you also specify that the OASIS interpolation times are provided directly by the lucia tool in table 8 captions (even if this is mentioned in the text on p.21)

**Answer:** The "computing time" measured by LUCIA and by the "time" function includes interpolation time. We introduced a clear analysis of extra cost, corrected the caption of table 8 and extended the caption of (now) Figure 6. The OASIS interpolation is now given clearly for each coupling.

My second concern is about the definition of the criteria to identify the optimum configuration, which are not clear:

In section 4.2, please specify what you mean by "each component's gain in speed, Compared to its speed on one node, outweighs the increase in costs." The units of speed are not the same as the units of cost so they cannot be compared directly. Are you considering the relative gain in speed (in %) is compared to the relative increase of cost (in %)? To help the understanding, one practical example with numbers should be given (maybe at the beginning of current section 4.4?), for example the steps that lead to the identification of the optimum configuration for the CCLM

**Answer:** Thank you for this comment. We agree that the description of the strategy was not sufficient to understand and reproduce the results presented. We revised (now) section 4.3 describing the strategy and give the numbers in the new section 4.4 describing the application of the strategy.

The optimum configuration is always a compromise between efficiency (depending on models scalability) and availability of resources or time to solution and cost. It is maybe not possible to give an objective definition of what this compromise should be. Thus we introduced a parameter for that compromise, the parallel efficiency: "The optimum configuration is found by starting the measuring of the computing time on one node for all components, doubling the resources and measuring the computing time again and again as long as all component parallel efficiencies remain above 50%. The threshold of 50% is subjective and can be defined by the user, i.e. one could decide to stop at a higher parallel efficiency if costs are a limiting factor." This definition is the same for both concurrent and sequential

configurations. An additional criterion is introduced, if the increase of cost has no impact on time to solution, in other words, if there is no scalability. In this case the parallel efficiency down to 50% is not used.

I think section 4.2 would be better to place just before 4.4 (i.e switching current 4.2 and 4.3 sections)

Answer: we follow the advice and switched the two §

p.22, I.749: This constraint is effective only for sequential coupling so it should be moved to the paragraph currently starting in line 754.

**Answer:** We have rewritten the sentence considering the reviewers suggestion (line 826 ff).

p.22 I.763 & I.779: It is not clear who or what decides if the costs are a limiting factor or not. Did you consider the costs was a limiting factor in your identification of the optimum? If so, what was the limit? This should be clarified.

**Answer:** Thank you for the comment. We rewrote the paragraph (line 836-849). We didn't introduce any other criterion but 50% parallel efficiency and, lowest cost, if no scalability is found. The application of the criteria is described in section 4.4 for each coupling in detail.

My third major concern is about section 4.5.2.I find this section not really relevant in the context of this paper. Of course, one can always get better fictive results by neglecting costly or badly written parts of the code!

**Answer:** We thank the reviewer for this suggestion. Section 4.5.2 is removed. Figure 6 and 7 are removed. Instead the extra cost of coupling for CCLM-MPIESM are discussed in section 4.5. line 984 ff and in section 4.6.

**Then I have the following major remarks:**

In general, I think the text is quite heavy with many repetitions. I suggest to make it lighter and more "right-to-the-point". In particular, section 2 describing the components could be reduced and the appendix A, that is not essential to the understanding of the paper could be given as supplementary material.

**Answer:** We would like to thank the reviewer for highlighting this important aspect of readability. The authors discussed this aspect again. Interestingly, reviewers 1 and 2 have dissenting opinions. You suggest reduction of the content and focusing on finding of an optimum configuration. The second reviewer suggests adding more details on configuration for asserting reproducibility and adding a discussion of the impact of configuration of model physics and dynamics on cost and time to solution. Considering the online publiccation form, we would like to keep section 2 in the paper. We revised the introductions of the sections in chapter 2 indicating that it is not essential for readers interested in the strategy of finding an "optimum configuration" only. The text is kept as it was with minor corrections. The introduction of the Appendix 1 is revised as well. It is essential for understanding of the coupling physics and dynamics and it is kept as appendix of the article since it does not increase the size of the PDF significantly and allows to keep everything in one document.

With which version of COSMO were the CCLM stand-alone tests done? Is it cosmo\_4.8\_clm19 like for all coupling but CCLM-CLM? This should be clarified in the text. **Answer**: Yes, see our answer below. We modified § 2.1, line156, and § 2.6 accordingly.

p.3, I.69: Please give some details on why the MESSy approach was not considered.

**Answer:** The CCLM couplings available with MESSy and OASIS are different. A comparison between MESSy and OASIS is planned for CCLM+MPI-ESM couplings. This requires additional developments for a fair comparison which are not finished yet.

p.5, I.597: You mention a "coupling weight" increasing to 1 with time but this coupling weight is not described. Can you explain with more details how it works? **Answer:** Thank you for this suggestion. The function used is given now in the text, line 601-605..

p.21, I.724: can you justify the formula used to approximate the time for 40 levels based on the time for 45 levels; why not simply use :  $T40 = T45 \times 40/45$

**Answer**: The scaling of 80% of the computing time (and not 100% as suggested by your comment) is already explained in the footnote.4, line 760.

p.24, 806-807: I do not understand how one can conclude that "COSMO-CLM inv ST and SMT mode exhibits a very similar PE for the same number of processes ..." The curves are distinct. Do you mean that we should compare the SMT results for a specific number of cores with the ST results with twice as many cores (to get to the same number of processes)?

**Answer:** Yes, this is what we wanted to say. We agree that the explanation is weakly understandable and improved it. See line 808 ff.

p.24, I.808: you write "an increased loss of PE between 160 and 80 grid points per process." but the reader cannot directly infer the number of grid points per process given the number of cores (which is the information provided on the figure), so the corresponding number of cores should be mentioned to help the reader.

**Answer**: We thank the reviewer for this comment and revised the paragraph for a better explanation. See line 814.

16. p.24, I.813-814: I do not fully understand this sentence. First I am not sure what the "component interface" is. Is it the coupling interface? If so, I do not understand how to reconcile this with the fact that the coupling interface time probably includes the time for interpolations (which are done either on the source side before the sending or on the target side after the receiving) and that the "time to solution" does not.

**Answer:** We thank the reviewer for raising this point. We agree that this sentence is not correct. We revised this paragraph and moved this explanation to "extra time and costs" in §4.5, line 1009.

p.24, I.815-819: I do not understand the meaning of the sentence "Hereby, the number of cores and the threading mode (ST or SMT) are kept constant." I propose to remove this sentence and rewrite the following ones as: " COSMO-CLM components of concurrent couplings should be compared to stand-alone COSMO-CLM in SMT mode because in both cases two threads per core are used to run COSMO-CLM. Conversely, COSMO-CLM components of sequential couplings should be compared to stand-alone thread per core is used to run COSMO-CLM." if I am right in my interpretation.

**Answer**: Thank you for this suggestion. The reference for each coupling is described now as suggested in section 4.4

p.24, I.826-827: It is written "However, as mentioned in section 2.6 CLM is coupled to cosmo\_5.0\_clm1 model version which is a more recent version than cosmo\_4.8\_clm19 used for all other couplings " but I don't see this mentioned in section 2.6.

**Answer:** We thank the reviewer for this remark. We corrected section 2.6 accordingly.. "The model version cosmo\_4.8\_clm19 is the recommended version of the CLM-Community (Kotlarski et al., 2014) and it is used as basis of the development of the couplings. CCLM as part of the CCLM+CLM coupled system is used in a slightly different version (cosmo\_5.0\_clm1). The way this affects the performance results is presented in section 4.5, line 954 ff. In addition, the reviewer can see, in the figure below, a scalability comparison between the 2 versions. This reveals (even though the machine is not the same than the one used in the article) the cost of the 5.0 version are 45% higher than for 4.8.

Figure 1: Time to solution of 5.0 and 4.8 COSMO-CLM versions in dependence on core number on Cray XC30 at CSCS, Lugano.

p.26, I.898-900: I propose to rephrase these two sentences for "It is not surprising that the couplings with soil-vegetation models shows only moderate extra costs as they replace the use of TERRA, the internal soil-vegetation model activated in stand-alone versions of COSMO-CLM."

**Answer**: We changed the sentences. They are now in § 4.5, line 953.

p.28, I.949-955: This paragraph is not clear. Going from non-alternating to alternating reduces the time to solution by 35.1 %. Improving the performances of the derivative calculation reduces the time to solution by 9.2%. Going from 16 cores in SMT mode to 32

cores in ST mode results in a reduction of time to solution by 25.5 %. But then why is the "discrepancy" calculated by comparing this 25.5% to the 9.2% linked to the improvement of the derivative calculations? It should be calculated by comparing the 25.5% to the non-alternating to alternating gain of 35.1%, shouldn't it?

**Answer**: Thank you for asking for clarification of this puzzling result. We explain this complex result now in more detail. See §4.6, line 1017 ff

p.30, I.1040-1043: The 10% variation in the time to solution results should be introduced in the text and not only in the conclusion.

Answer: Thank you for the suggestion. We added this information at the end of §4.1

**Minor remarks and technical corrections**

I think it would be less confusing to use CCLM everywhere and not sometimes CCLM and sometimes COSMO-CLM

**Answer**: COSMO-CLM is the official name chosen by the CLM community, with CCLM as the official abbreviation when there is not enough space like in figures. We use now CCLM nearly everywhere. However, to avoid confusion between CLM and CCLM, the full name COSMO-CLM is used more than once.

p.1, I.8: The OASIS3-MCT interface is not really described in the paper. I suggest changing "present" for "use".

**Answer**: Thank you for the comment. We realised that we introduced a confusion by using "interface" for model routines where coupling is performed instead for the OASIS3-MCT API (widely known as "PSMILE library"). We modified the text: "We present a unified interface, based on OASIS3-MCT coupling library"

p.3, I.58 & p.6, I.166: Valcke 2013 refers to a paper describing the "old" OASIS3 version and not the more recent OASIS3-MCT version. The reference Valcke et al., 2013 should be used instead.

Answer: We changed the reference.

p.3, I.67: I propose changing "is based" for "would be based" **Answer**: We changed the text, I67.

p. 4, I.94: Please add "depends" after "but" in "but on the coupling method **Answer:** Done

p.6, I.168: Please add "which" after "data" in "amount of data is a requirement" **Answer:** Done

p.6, I.186-187: The sentence "The coupling of COSMO-CLM with the global ocean model NEMO is realized by means of two different regional versions of the NEMO model ..." sounds weird to me because of the opposition between "global" and "re- gional". I suggest simply "COSMO-CLM is coupled to two different regional versions of the NEMO model ..." **Answer:** Done, line 191.

p.10, I.337: The fact that each component needs to be a separate executable is not a constraint anymore with the last OASIS3-MCT\_3.0 version; maybe this could be mentioned.

**Answer**: We added a remark on that feature in OASIS3, See line 340.

p.11, I.339 & I.366: Please change "whose" for "which"

**Answer: Done, line 344 and 373**

p.11, I.343: I suggest changing "is directly executed via the Message Passing Interface" for "is directly executed via the Model Coupling Toolkit (MCT, Jacob et al 2005) based on the Message Passing Interface (MPI)" and add the reference "Jacob, R., J. Larson, and E. Ong: MxN Communication and Parallel Interpolation in CCSM3 Using the Model Coupling Toolkit. Int. J. High Perf. Comp. App., 19(3), 293-307 2005 "

**Answer**: We thank the reviewer for this suggestion making this point more clear. We changed the text accordingly, line 349 ff.

P.11, I.357-358: I suggest changing "This component partitioning does not have to be the same" for "The component partitioning and grid do not have to be the same" **Answer**: We thank the reviewer for this suggestion and changed the text accordingly in line 364.

p.11, I.361: I suggest adding "and accumulation" after "time averages" I propose "average or accumulation" **Answer**: Done, line 368

p.11, I.373: I suggest changing "OASIS3-MCT includes the MPI library" for "OASIS3-MCT includes the MCT library based on MPI" (but this is redundant with p.11, I.343 -see also my remark #9 above

**Answer**: We followed the reviewers suggestion and keep the redundancy for better readability, line 380.

p.13, I.428: Please add a ) after 4.1 **Answer:** Done, line 435.

p.13, I. 442: Please change "interpolation" for "coupling" as it is not only the interpolation that is improved but the interpolation and the communication. **Answer**: We added "and communication" for clarity, line 449.

p.20, I.687-691: I am not convinced these are effectively the two main goal of performance analysis. These sentences are unnecessary and contribute to the heaviness of the text (see also my first "Important remark" above.

**Answer**: Thank you for the comment. We removed the discussion of what is not done and changed the text accordingly. See line 718 ff.

p.21, I.722: Please change "compansated" for "compensated" **Answer**: Done, line 761.

p.22, I.737-738, I suggest rephrasing the sentence "In a perfectly scaling parallel application the costs would remain constant if the resources are doubled, the parallel efficiency would be 100 %, the speed would be doubled and the speed-up would be 200 %. " for "If the resources of a perfectly scaling parallel application are doubled, the speed would be doubled and therefore the cost would remain constant, the parallel efficiency would be 100 %, and the speed-up would be 200 %."

**Answer**: We thank the reviewer for this suggestion and changed the text accordingly, line 777.

p.23, I.791: Please change "CPUh" for "core hours" to be coherent with the rest of the text.

Table 8 should be placed after Table 7 and not after all the Figures. **Answer**: We thank the reviewer for this suggestion. Table 8 is now located after figure 6 showing the optimum configurations.

p.26, I906: Please change "atmosphere" for "coupled model" **Answer**: We thank the reviewer for this suggestion and changed the text accordingly, line 991 ff.

p.30, I.1031: Please change "scaling;" by "scaling,"

**Answer**: We thank the reviewer for this suggestion and changed the text accordingly, line 1077.

---

## Author Response (AR2)

**Author Response to "Topical Editor Decision: Publish subject to technical corrections (04 Feb 2017)"**

Andreas Will and colleagues

February 12, 2017

Cottbus, 12 Feb. 2017

**Dear Dr. Fyke**,

we would like to thank you for your carefull reading of the revised manuscript and the suggestions and comments you made. The 2nd revised version (5.12 is our number) uploaded consideres all of them and we hope that it improved the understandability of the results, the consistency of the presentation and the readability.

We also tried to improve the language as much as possible. However, wa are non-native speakers english.

You may find our answers to your detailed comments in the following.

Wer are looking forward to your final decision and remain with kind regards, Andreas Will and co-authors

**Ms. Jeremy Fyke's review of the revised manuscript (4 Feb 2017) and author response:**

Thank you for providing a revised manuscript. In reviewing it, I have found it quite dense and disjointed (even for a model description paper). Hopefully we can improve it prior to publication, so it will be accessible and helpful to readers. To that end I have the following suggestions for technical corrections prior to publication:

1. Improve English and conciseness of Abstract and Introduction Answer: We revised the abstract, introduction and conclusions and hope that we improved the language and readability.

2. better differentiation of COSMO-CLM and Community Land Model (CLM) acronyms. Quite confusing for the reader
   Answer: We fear, this problem cannot be solved. Both models have their acronyms for more than 10 years and it is not wise to introduce a new acronym in this article. Usually COSMO-CLM is used in the text and CCLM in figure captions. However, here we need to

use CCLM+VEG3D etc. and we agree that using both COSMO-CLM and CCLM in the text is even more confusing. Thus we removed COSMO-CLM everywhere but at positions at which the full name is used "COSMO model in Climate Mode". We hope, it's the best solution possible.

3. A more comprehensive tie-together of all the models used. Currently, to the fist-time reader it appears as somewhat of a laundry list. For example:

   - better high-level justification of reasoning for tying together so many disparate models
   - Justification for the couplings to different ocean models, depending on basin
   - A better model-specific justification for why each model was chosen

   Answer: Thank you for the suggestion. We revised the paragraph introducing the individual couplings (line 56 ff.) accordingly. Furthermore, the abstract is rewritten and the conclusions are substantially revised.

4. is the full MPI-ESM coupled model is used, or just the atmosphere? The first time reader is left confused as to the explicit connections between this coarse global model, and COSMO-CLM, which are only revealed as 2-way coupled, deep in the technical descriptions (unless I missed something)
   Answer: Thank you for this hint. We removed the ambiguity and explain the general aspects of the coupling in the introduction. We explain now the method of coupling (atmosphere-atmosphere) between CCLM and MPI-ESM in the interface description only.

5. "An overview of the coupled models selected for coupling with COSMO-CLM (CCLM) is given in table 3": the term coupled models implies they already all coupled. It is thus confusing to read that they will be coupled *again* to something else.
   Answer: We agree that the wording is confusing. We replaced "coupled models" by "models coupled (with CCLM)" throughout the text or replaced it by *components* of the coupled system throughout the text. Furthermore, we also replaced "model components" and "component models" by *components*. We hope this improves the readability.

6. confusing: "atmosphere-atmosphere coupling" or "ocean-ocean coupling". Need to be more clear that it is globalTOregional one/two way coupling (if I understand correctly).
   Answer: Thank you for the hint. We removed this wording from the introduction and highlight in the introduction now that all couplings are between the regional climate model CCLM and another model and leave the details of the coupling method for the interface description section.

7. It is confusing that some models being coupled to COSMO are component models (e.g. CLM), while MPI-ESM is a full coupled model. A clearer up-front description of this differentiation (if correct) is necessary.
   Answer: We agree that MPI-ESM is an earth system model while e.g. NEMO or CLM are models of one component of the earth system, i.e. component models. We follow the

wording in OASIS3-MCT and name all models coupled now "components" (of the coupled system throughout the paper.

8. The climatological means of freshwater inflow of 33 rivers to the North Sea and the Baltic Sea are collected from Wikipedia. A more robust source is needed here. Answer: We removed the citation of Wikipedia. Instead we give a more precise description of the numbers used and cite the author of that approach as personal communication. Unfortunately, there is no more precise citation available.

9. Check that all acronyms are first defined in the text, and then the acronyms are only used in the future. For example, COSMO-CLM is defined as "CCLM", but then "COSMO-CLM" is used again in full, later. Answer: Thank you for this hint. We replaced now COSMO-CLM and COSMO by CCLM throughout the text.

10. There are two sections, titled "OASIS3-MCT coupling method and performance optimization", followed by "OASIS3-MCT coupling method". These titles are redundant, and I suggest a careful merge and re-write of these sections into one coherent section. Answer: We followed the suggestion and merged the sections.

11. It is not clear to me early in the manuscript, whether MPI-ESM is used only as generators of boundary conditions for the regional model, or if two-way coupled has been implemented between these. Please clarify early on, so the readers are prepared for the later, more detailed descriptions. Answer: Thank you for the hint. Throughout the paper we investigate a two-way coupling between MPI-ESM and CCLM. We revised the description of the coupling CCLM+MPI-ESM in the introduction (line 71 ff) and at other places of the article to clarify this point.

12. "Here `send_fld` ends": what is `send_fld`? Answer: Thank you for this question. `send_fld` is a coupling routine in CCLM. We revised section 3.2 by adding the references to the time-step organisation figure and gave a more precise overview of what is done in which routine.

13. Has the two-way-coupled CCLM+MPI-ESM system actually been prognostically run (as opposed to using a prescribed COSMO-CLM solution)? Answer: Thank you for this question. We added a sentence in the introduction clarifying that all coupled systems presented have been used successfully on climatological time scales. For CCLM+MPI-ESM we added this comment in 3.2 as well.

14. In Conclusions: perhaps make a more 'tied together´ set of conclusions, from the many paragraphs that start with the words: "The optimum configuration of the coupling". Or alternatively, make bullet points. In either case, the optimal coupling strategies should be more accessible to readers, who are potentially interested in each individual coupling. Answer: Thank you for the suggestion. We revised the conclusions accordingly.

15. In Conclusions: I think it is necessary in the concluding section, to return to the initial scientific reasons for constructing COSMO-CLM couplings in the first place. This will

remind the reader why this technical work is relevant to a broader research efforts, and perhaps motivate the reader to work with these couplings themselves. Answer: Thank you for this suggestion as well. We revised the fist paragraph of the conclusions accordingly.